# Dynamic viability of the 2016 Mw 7.8 Kaikōura earthquake cascade on weak crustal faults

Thomas Ulrich[1], Alice-Agnes Gabriel [1], Jean-Paul Ampuero [2,3] & Wenbin Xu [4]

We present a dynamic rupture model of the 2016 $M_w$ 7.8 Kaikōura earthquake to unravel the event's riddles in a physics-based manner and provide insight on the mechanical viability of competing hypotheses proposed to explain them. Our model reproduces key characteristics of the event and constraints puzzling features inferred from high-quality observations including a large gap separating surface rupture traces, the possibility of significant slip on the subduction interface, the non-rupture of the Hope fault, and slow apparent rupture speed. We show that the observed rupture cascade is dynamically consistent with regional stress estimates and a crustal fault network geometry inferred from seismic and geodetic data. We propose that the complex fault system operates at low apparent friction thanks to the combined effects of overpressurized fluids, low dynamic friction and stress concentrations induced by deep fault creep.

[1] Department of Earth and Environmental Sciences, Ludwig-Maximilians-Universität, 80333 München, Germany. [2] Géoazur, Université Côte d'Azur, IRD, CNRS, Observatoire de la Côte d'Azur, 06560 Valbonne, France. [3] Seismological Laboratory, California Institute of Technology, Pasadena, CA 91125-2100, USA. [4] Department of Land Surveying and Geo-Informatics, Hong Kong Polytechnic University, 999077 Hong Kong, China. Correspondence and requests for materials should be addressed to T.U. (email: ulrich@geophysik.uni-muenchen.de)

The $M_w$ 7.8 Kaikōura earthquake struck New Zealand's South Island on November 14, 2016. This event, considered the most complex rupture observed to date, caused surface rupture of at least 21 segments of the Marlborough fault system, some of them previously unknown. Here we develop a dynamic rupture model to unravel the event's riddles in a physics-based manner. Our model reproduces strike-slip and thrust faulting and requires a linking low-dipping shallow thrust fault, but not slip on an underlying megathrust. The apparent rupture slowness is explained by a zigzagged propagation path and rupture delays at the transitions between faults. The complex fault system operates at low apparent friction owing to the combined effects of overpressurized fluids, low dynamic friction, and stress concentrations induced by deep fault creep. Our results associate the non-rupture of the Hope fault, one of the fundamental riddles of the event, with unfavorable dynamic stresses on the restraining step-over formed by the Conway-Charwell and Hope faults.

Studies of the Kaikōura earthquake based on geological, geodetic, tsunami, and seismic data reveal puzzling features as well as observational difficulties. An apparent gap of 15–20 km in surface rupture between known faults[1] may suggest a rupture jump over an unexpectedly large distance or the presence of deep fault segments connecting surface rupturing faults. Rupture duration is long, more than twice the average duration of past earthquakes of comparable magnitude[2]. Finite-fault source inversion models inferred from strong motion and other data[3–5] present unconventional kinematic features, such as unusually large delays between segments[3] or strong scatter in the distribution of rupture time[5]. The rupture may include simultaneous slip on the Hikurangi subduction interface[5] and several segments slipping more than once[4]. Teleseismic back-projection studies[6–8] agree on general earthquake characteristics (e.g., an overall SW-NE propagation direction) but not on the space–time evolution of the rupture.

Competing views of the role played by the Hikurangi subduction interface during the Kaikōura earthquake have emerged from previous studies. Whereas far-field teleseismic and some tsunami data inferences require thrust faulting on a low-dipping fault, interpreted as the subduction interface beneath the Upper Kowhai and Jordan Thrust faults[2,5,6,9], analysis of strong motion, aftershocks, geodetic, and coastal deformation observations find little or no contribution of the subduction interface[4,7,10,11]. The geometry of the Hikurangi megathrust is not well constrained in its Southern end[12]: dipping angles assumed in previous studies range from 12° to 25°[1,5]. Large-scale ground deformations have then been explained by either slip on the subduction interface (e.g., refs. [1,5]) or by crustal models featuring listric fault geometries[7] or shallow thrust faults[10].

Incorporating the requirement that the rupture should be dynamically viable can help constrain the unexpected features and competing views of this event. Analyses of static Coulomb failure stress changes during rupture provide some mechanical insight on the rupture sequence[1,7], but do not account for dynamic stress changes, which are an important factor in multi-fault ruptures (e.g., ref. [13]). Dynamic rupture simulations provide physically self-consistent earthquake source descriptions, and have been used to study fundamental aspects of earthquake physics (e.g., refs. [14,15]), to assess earthquake hazard (e.g. ref. [16]) and to understand previous earthquakes (e.g., refs. [17,18]). The dynamic rupture modeling presented here provides physical arguments to discriminate between competing models of the fault system geometry and faulting mechanisms.

Mature plate boundary faults are, in general, apparently weak[19–21], a feature that is required also by long-term geodynamic processes (e.g., refs. [22,23]), but that seems incompatible with the high static frictional strength of rocks[24]. These two observations can be reconciled by considering dynamic weakening, which allows faults to operate at low average shear stress[25]. However, low background stresses are generally unfavorable for rupture cascading across a network of faults. For instance, rupture jumps across fault step-overs are hindered by low initial stresses[13]. This is one reason why finding a viable dynamic rupture model is non-trivial. The modeled fault system presented here features a low apparent friction while being overall favorably oriented with respect to the background stress. We demonstrate that fault weakness is compatible with a multi-fault cascading rupture. Our models suggest that such a weak-fault state is actually required to reproduce the Kaikōura cascade (see "Apparent fault weakness" in Methods section).

Our dynamic model of the Kaikōura earthquake is tightly determined by integrating knowledge and data spanning a broad range of scales. It combines an unprecedented degree of realism, including a modern laboratory-based friction law, off-fault inelasticity, seismological estimates of regional stress, a realistic fault network geometry model, a three-dimensional (3D) subsurface velocity model, and high-resolution topography and bathymetry. High-resolution 3D modeling is enabled by the SeisSol software package that couples seismic wave propagation with frictional fault failure and off-fault inelasticity, and is optimized for high-performance computing (see "Numerical method" in Methods section).

The resulting dynamic model of the Kaikōura earthquake sheds light on the physical mechanisms of cascading ruptures in complex fault systems. Our model reproduces key characteristics of the event and constraints puzzling features including a large gap separating surface rupture traces, the possibility of significant slip on the subduction interface, the non-rupture of the Hope fault, and slow apparent rupture speed. We show that the observed rupture cascade is dynamically consistent with regional stress estimates and a crustal fault network geometry inferred from seismic and geodetic data under the assumption of low apparent friction.

## Results

**Fault geometry**. We construct a model of the non-planar, intersecting network of crustal faults (Fig. 1) by combining constraints from previous observational studies and from dynamic rupture modeling experiments. Fault geometries and orientations have been constrained by geological and geodetic data (e.g. refs. [7,26,27]). Our starting point is a smoothed version of the fault network geometry model III inferred from field and remote sensing data by Xu et al[7]. It comprises three strike-slip faults: Humps and Stone Jug faults and a long segment with listric geometry (flattening at depth) resembling jointly the Hope-Upper Kowhai-Jordan Thrust, Kekerengu, and Needles faults; and four thrust faults: Conwell-Charwell, Hundalee, Point Kean, and Papatea faults. The model does not include the subduction interface, but is sufficient to explain the observed static ground deformations in the near field and far field.

We extend this simplified model to capture the complexity of the southern part of the fault network. The western tip of the Humps segment is slightly rotated (azimuth direction from WSW to W) in our model. The improved agreement with the mapped surface rupture enables spontaneous termination of the westward rupture front. We substitute the Conway-Charwell fault zone by the distinct Leader and Conway-Charwell faults[27]. The geometry of the Leader fault is similar to the Conway-Charwell fault zone of Xu et al.'s[7] model; however, the former is increasingly steeper to the North. Surface rupture mapping suggests a segmentation of the Leader fault in at least two segments[27]. Yet, the continuity of

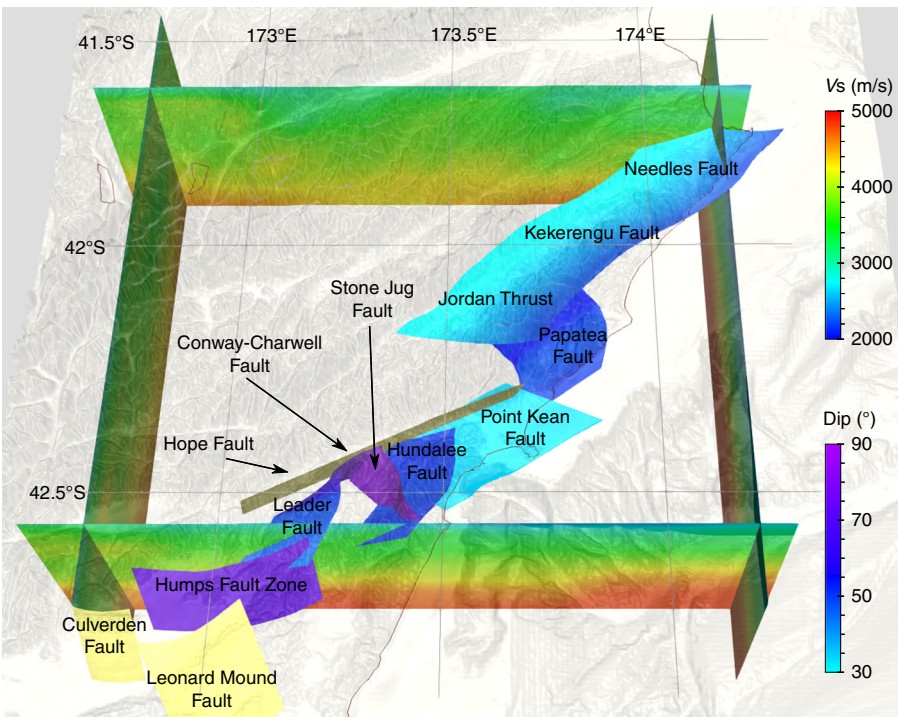

**Fig. 1** Fault network geometry prescribed for dynamic earthquake rupture modeling. Colors on fault surfaces indicate dipping angle (dip), highlighting the flattening with depth of the Jordan Thrust, Kekerengu, and Needles faults. All segments dip westwards, except for the Humps Fault Zone. The Hope, Culverden, and Leonard Mound faults, dipping, respectively, 70° toward NorthWest, 70° toward South, and 50° toward SouthEast, are displayed in yellow. These faults do not rupture in our dynamic rupture model. Also shown are the high-resolution topography and bathymetry[58], and S-wave speeds ($V_s$) on four cross-sections of the 3D subsurface structure[59] incorporated in the model

the inferred ground deformations in that region[27] suggests a unified segment. Dynamic rupture experiments accounting for a large step-over within the Leader fault also suggest that a segmented geometry is not viable. The Conway-Charwell fault steps over the Leader fault. It runs roughly parallel to the Hope fault to the North. The Southernmost part of the long listric segment of Xu et al.'s[7] geometry, representing the Hope fault, is replaced here by the Hope fault geometry proposed by Hamling et al.[1], which is more consistent with the mapped fault trace and inferred dip angle[26]. The 60° dipping Stone Jug fault of Xu et al.[7] is replaced by a steeper fault, as suggested by Nicol et al.[27] The Hundalee segment is shortened at its extremities, to limit its slip extent according to Xu et al.'s[7] inversion results.

Based on experimental dynamic rupture simulations, we remove the Upper Kowhai fault. Instead, we postulate that the previously unknown Point Kean fault[10] acted as a crucial link between the Hundalee fault and the Northern faults. The Upper Kowhai fault is well oriented relative to the regional stress and, when included, experiences considerable slip in contradiction with observations. Although geodetic data suggest a moderate amount of slip on this fault at depth[1,7], we hypothesize that such slip is not crucial for the continuation of the main rupture process. This is supported by recent evidence suggesting the rupture propagated from the Papatea fault to the Jordan thrust (more details in "Strong ground-motion and continuous Global Positioning System (GPS) data" in Results section), rather than a Jordan thrust–Papatea fault sequence mediated by slip on the Upper Kowhai fault. Moreover, localized slip at depth on the Upper Kowhai fault would be difficult to reproduce without additional small-scale features in the fault geometry or fault strength heterogeneities.

**Friction**. We constrain our model parameters based on findings from laboratory to tectonic scale. Specifically, incorporating

realistic levels of static and dynamic frictional resistance and stress drop is an important goal in our model design.

In our model, adopting a friction law with severe velocity weakening enables full cascading rupture and realistic amounts of slip, in contrast to simplified friction laws. We adopt a friction law featuring rapid weakening at high slip velocity (adapted from Dunham et al.[28] as detailed in "Fault friction" in Methods section), which reproduces the dramatic friction decrease observed in laboratory experiments at co-seismic slip rates[29]. Compared to results of our numerical experiments with linear slip-weakening friction (e.g., ref.[30]) on the same fault geometry, we find that strong velocity-weakening facilitates rupture cascading because it yields a smaller critical size to initiate self-sustained rupture by dynamic triggering.

**Initial stresses**. The stresses acting on natural faults and their strength are difficult to quantify. Although strength parameters are measured in laboratory friction experiments[29] and estimated from different types of observations[31], little consensus about the actual strength of faults exists[32]. We introduce new procedures to constrain the initial fault stress and relative strength. This systematic approach, detailed in "Initial stresses" in the Methods section and Supplementary Fig. 1, is constrained by observations and simple theoretical analysis, including seismo-tectonic observations, fault slip inversion models, deep aseismic creep, fault fluid pressurization, Mohr–Coulomb theory of frictional failure, and strong dynamic weakening. In addition to static analysis, it requires only few trial simulations to ensure sustained rupture propagation. By efficiently reducing the non-uniqueness in dynamic modeling, this approach is superior to the common trial-and-error approach.

Our initial stress model is fully described by seven independent parameters (Supplementary Fig. 1): four parameters related to regional stress and seismogenic depth, which are directly

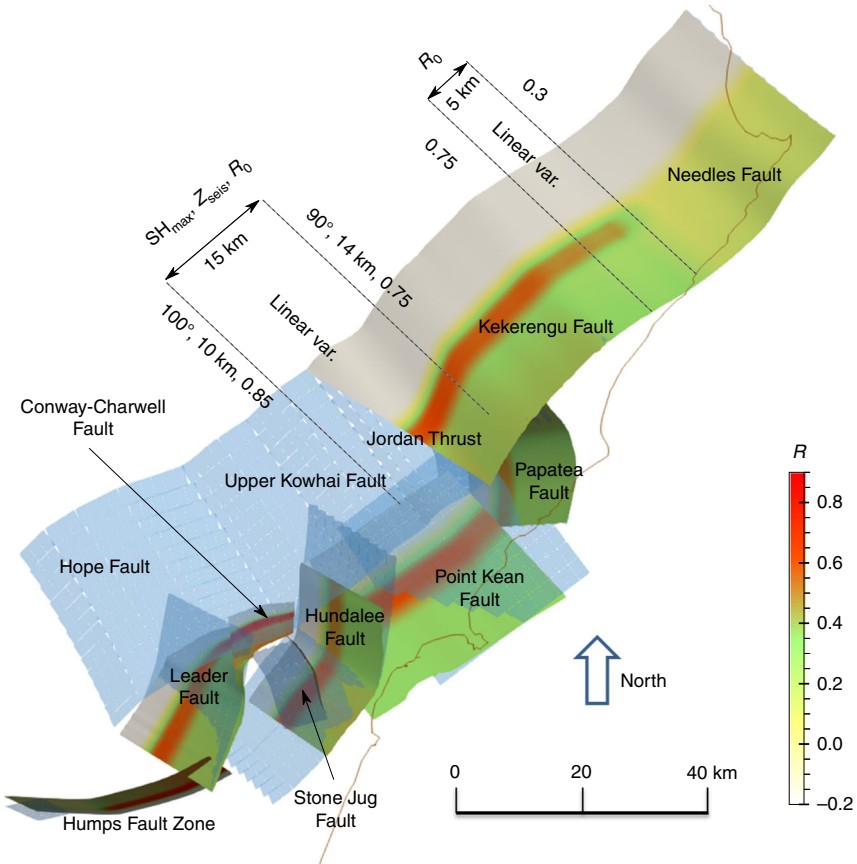

**Fig. 2** Adaptations made to the geodetically inferred fault geometry to develop a realistic dynamic rupture model. Changes made to the fault geometry are highlighted by plotting the here used geometry over the fault geometry of Xu et al.[7] shown in transparent blue. The distribution of initial fault stress ratio $R$ (Eq. 10) along the fault network is also shown. The spatial distribution of parameters defining the stress ($SH_{max}$, $R$, and $z_{seis}$ defined in "Initial stresses" in Methods section) are indicated. The magnitude of the initial stress loading is decreased in the Needle fault region to prevent large slip on this optimally oriented segment (such a decrease is modeled by decreasing $R_0$ by 60% and suppressing the deep stress concentrations in that region)

constrained by observations, and three unknown parameters related to fluid pressure, background shear stress, and the intensity of deep stress concentration. A stress state is fully defined by its principal stress magnitudes and orientations. The orientation of all components and the relative magnitude of the intermediate principal stress are constrained by seismological observations[33]. In addition, the smallest and largest principal stress components are constrained by prescribing the prestress relatively to strength drop on optimally oriented fault planes[34]. To determine the preferred initial stresses, we first ensure compatibility of the stress state with the prescribed fault geometry and the slip rakes inferred from static source inversion. In this purely static step, we determine optimal stress parameters, within their identified uncertainties, that maximize the ratio of shear to normal stress and maximize the alignment between fault shear tractions and inferred slip[7]. We then use a set of dynamic rupture simulations to determine the depth-dependent initial shear stress and fluid pressure that lead to subshear rupture and slip amounts consistent with previous source inversion studies. The resulting model incorporates overpressurized fault zone fluids[35–37] with a fluid pressure considerably higher than hydrostatic stress but well below lithostatic level (see "Initial stresses" in Methods section).

A favorable stress orientation on all segments, including thrust and strike-slip faults, is promoted by an intermediate principal stress close to the minimum principal stress[38] representing a transpressional regime. This configuration promotes thrust faulting on faults dipping at ~60° and striking perpendicularly

to the direction of maximum compression, which roughly corresponds with the thrust fault geometries of our model.

In our model, dynamic rupture cascading is facilitated by deep stress concentrations (Fig. 2). The presence of stress concentrations at depth near the rheological transition between the locked and steady sliding portions of a fault is a known mathematical result of the theory of dislocations in elastic media (e.g. refs. [39,40]). Such stress concentrations are also a typical result of interseismic stress calculations based on geodetically derived coupling maps[41] or long-term slip rates[42]. Stress concentrations due to deep creep on the megathrust have been proposed to determine the rupture path independent of crustal fault characteristics[43]. Stress concentration is introduced in our model by two independent modulation functions (Supplementary Fig. 2).

Our initial stress model leads to low values of the initial shear to normal stress ratio over most of the seismogenic zone (the median value over the rupture area is 0.09, see Supplementary Fig. 3) in consistence with the apparent weakness of faults[31] (see "Apparent fault weakness" in Methods section). Yet, most faults of our model are relatively well oriented with respect to the regional stress, and are therefore not weak in the classical sense. The classical Andersonian theory of faulting may be challenged in transpressional tectonic stress regimes resulting in non-unique faulting mechanisms. In the framework of dynamic rupture modeling, faults can be stressed well below failure almost everywhere and yet break spontaneously if triggered by a small highly stressed patch. Under the assumption of severe

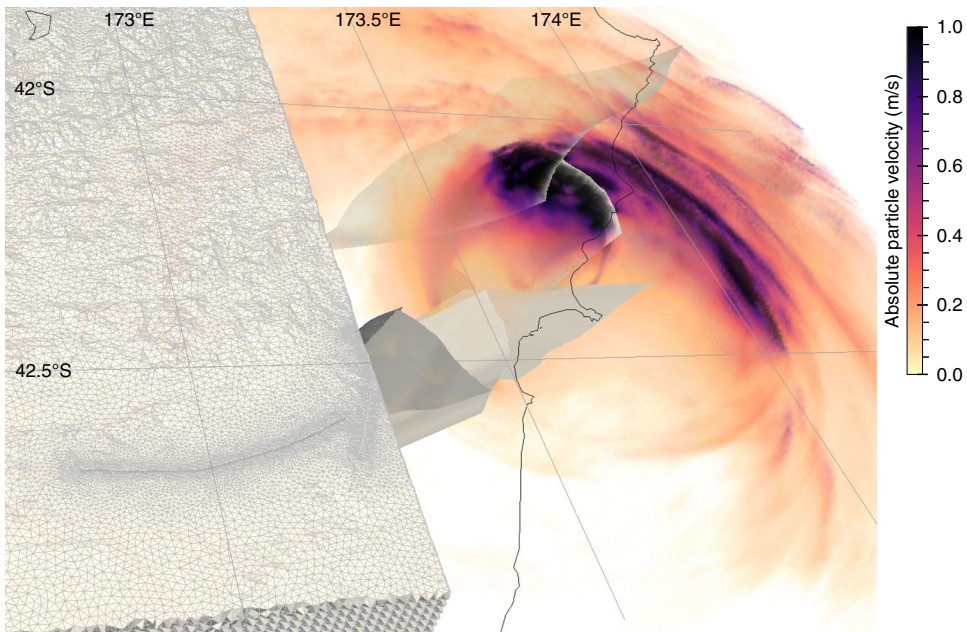

**Fig. 3** Snapshot of the wavefield (absolute particle velocity in m/s) across the fault network at a rupture time of 55 s. The model is discretized by an unstructured mesh accounting for three-dimensional (3D) subsurface structure and high-resolution topography and featuring refined resolution in the vicinity of the faults. It incorporates the nonlinear interaction between frictional on-fault failure, off-fault plasticity, and wave propagation

velocity-weakening friction (detailed in the previous section), a low level of prestress is required to achieve a reasonable stress drop. To this end, we have considered here two effects rarely taken into account together in dynamic rupture scenarios: (1) increased fluid pressure and (2) deep stress concentrations. We discuss their trade-offs in more detail in "Initial stresses" in the Methods section. We infer that the interplay of deep creep, elevated fluid pressure and frictional dynamic weakening govern the apparent strength of faults and that these factors cannot be treated in isolation for such complex fault systems.

Further minor adjustments of the initial stresses are motivated by observations. To prevent excessive thrust faulting of the Kekerengu fault, we introduce a rotation of the maximum compressive stress orientation, within its range of uncertainty, from 100° in the South to 90° in the North. We also introduce a North–South increase of the seismogenic depth to allow deeper slip on the Papatea and Kekerengu faults, and a slight decrease of initial stress magnitude. Collectively both measures improve the model agreement with observed far-field ground deformations and rupture speed (they prevent shallow supershear rupture). Finally, we locally reduce the initial stresses on the Northernmost part of the Needles fault to prevent the occurrence of large slip in this area. We find that the Needles fault would otherwise host more than 10 m of slip, which is not supported by inversion results[1,7].

**A dynamically viable, cascading rupture**. In our dynamic model rupture propagates spontaneously across eight fault segments (Fig. 1, which also shows three non-activated fault segments). The combined rupture length exceeds 240 km. The rupture successively cascades from South to North, directly branching at variable depths from the Humps to the Leader, Conwell-Charwell, Stone Jug, Hundalee, and Point Kean faults. It then jumps to the Papatea fault via dynamic triggering at shallow depth, and finally branches to the Jordan Thrust (Fig. 3), Kekerengu, and Needles faults (Fig. 4). This rupture cascade is dynamically viable without slip on the underlying subduction interface.

**Fault slip**. The modeled slip distributions and orientations are in agreement with the existing results[7,10]. We observe an alternation of right-lateral strike-slip faulting (Humps, Conwell-Charwell, Jordan Thrust, Kekerengu, and Needles faults) and thrusting (Leader, Hundalee, and Papatea faults), as well as left-lateral strike-slip rupture of the Stone Jug fault and oblique faulting of the Point Kean fault (Fig. 5). Due to the smoothness of our assumed initial stresses, the final slip distribution is less patchy than in source inversion models. However, the moment magnitude of 7.9 is in excellent agreement with observations (Fig. 5f).

**Apparent rupture speed**. The complexity of the rupture cascade contributes to its apparently slow rupture speed. The ratio of rupture length to rupture duration (inferred from moment rate functions estimated by various authors; Fig. 5f, refs. [8,9,44]) indicates a slow average rupture velocity of about 1.4 km/s [7]. In our model, rupture along each segment propagates twice as fast, at 2.9 km/s on average. Nevertheless, the observed rupture duration of approximately 90 s is reproduced owing to a zigzagged propagation path accompanied by rupture delays at the transitions between segments (see Supplementary movies 1 and 2). Specifically, the modeled rupture sequence takes about 30 s to reach the Hundalee fault after nucleation, whereas a hypothetical, uninterrupted rupture propagating at a constant speed of 3 km/s from the Humps to Hundalee faults would take only half this duration. The geometrical segmentation of the Leader and Conway-Charwell faults delays rupture by more than 5 s. Rupture across the Conway-Charwell fault is initiated at shallow depth. The Stone Jug fault can subsequently only be activated after rupture reached the deep stress concentration area and unleashed its triggering potential, causing further delay.

**Moment release**. Specific episodes of the dynamic rupture model can be associated to prominent phases of moment release and high-frequency radiation observed in the Kaikōura earthquake. Abrupt changes in rupture velocity during the entangled Leader–Charwell-Conwell–Stone Jug fault transition 20 s after rupture onset may correspond to a burst of high-frequency energy[45]

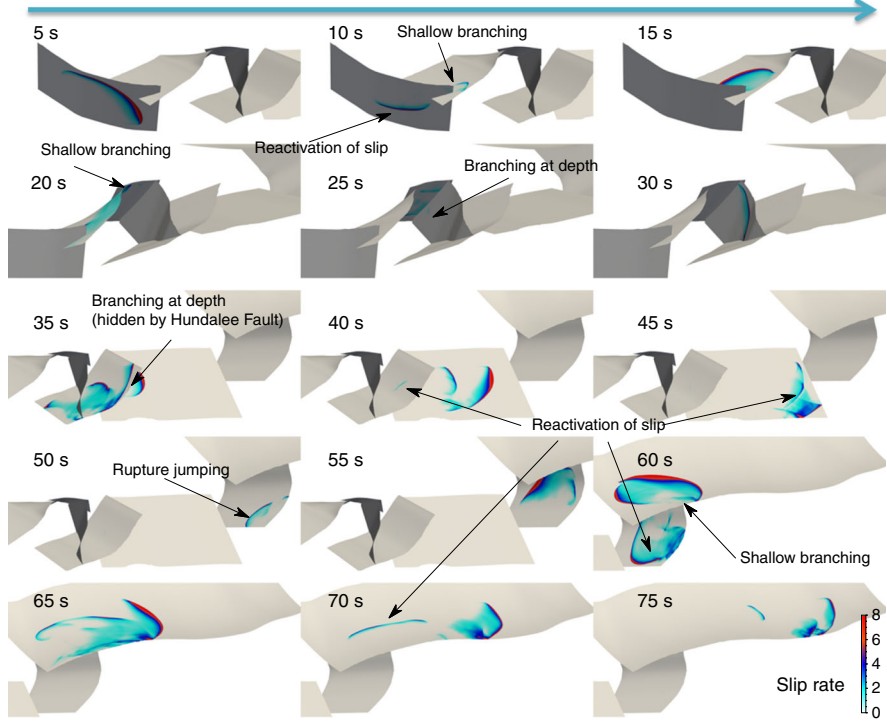

**Fig. 4** Overview of the simulated rupture propagation. Snapshots of the absolute slip rate are shown every 5 s. The figure focuses on four different portions of the fault system, following the rupture front as it propagates from South to North. Labels indicate remarkable features of the rupture discussed in the text

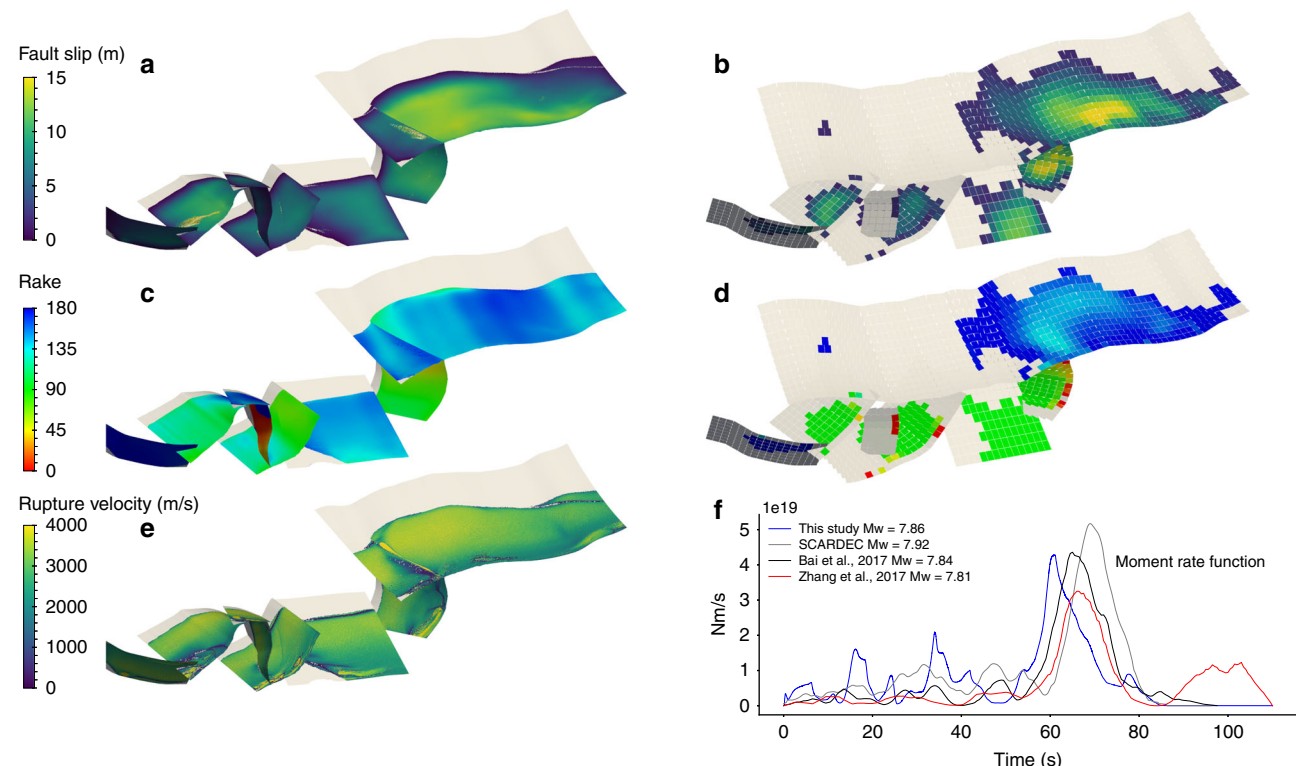

**Fig. 5** Source properties of the dynamic rupture model and comparison to observational inferences. Final slip magnitude **a** modeled here and **b** inferred by Xu et al.[7] Final rake angle **c** modeled and **d** inferred by Xu et al.[7] **e** Modeled rupture velocity. **f** Modeled moment rate function compared with those inferred by Bai et al.[9] from teleseismic and tsunami data, by Zhang et al.[8] from seismic waveform inversion and from teleseismic data by the SCARDEC method[44]

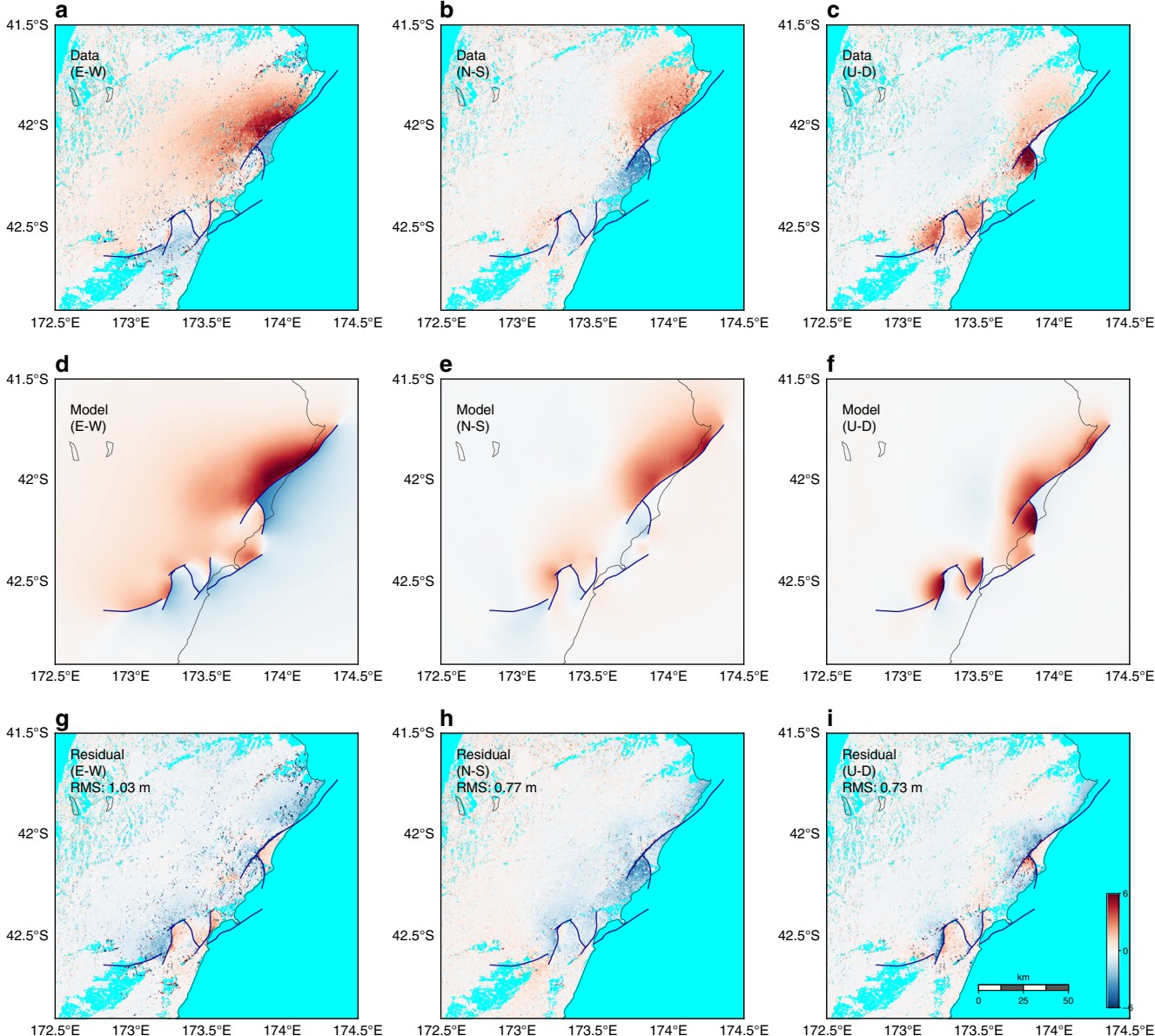

**Fig. 6** Comparison of observed and modeled co-seismic surface displacements. Three-dimensional (3D) ground displacement (first row, **a–c**) inferred by space geodetic data[7], (second row, **d–f**) generated by the dynamic rupture model and (third row, **g–i**) their difference, all in meters. Columns from left to right are EW, NS, and UD components. Root-mean-square (RMS) misfits are provided in the third row for each component

noted by back-projection studies[7,8]. Around 60 s after rupture onset, a distinct moment release burst lasting 20 s corresponds to the simultaneous failure of the Papatea and Kekerengu faults and is well aligned with observations[8,9,44].

**Ground deformation**. The static ground deformation in our model is in good agreement with that inferred from geodetic data[1,7] (Figs. 6 and 7). In particular, the maximum horizontal deformation along the Kekerengu fault and the substantial uplift near the intersection between the Papatea and Kekerengu faults are captured, and the observed ground deformation near the epicenter is reasonably replicated.

**Strong ground-motion and continuous GPS data**. Strong ground-motion and continuous GPS data provide valuable constraints on the rupture kinematics. We compare our simulation results to these data with a focus on the timing of pulses, because

our model does not account for small-scale heterogeneities that could significantly modulate waveforms. Due to the close distance of some of the stations to the faults (Fig. 8), a close match of synthetic and observed waveforms is not expected. Yet, the dynamic rupture model is able to reproduce key features of the strong ground-motion and GPS recordings (Fig. 9). Our model captures the shape and amplitude of some pronounced waveform pulses, for example, of the first strong pulse recorded along the NS direction at GPS station MRBL, which is situated in the nucleation area. A time shift of around 2 s hints at a nucleation process slower than modeled. At near-fault station KEKS two dominant phases are visible on both observed and synthetics waveforms (at 52 and 63 s after rupture onset in the NS synthetics of Fig. 9 and in the fault-parallel-rotated waveforms of Supplementary Fig. 4). These dominant phases were attributed to a slip reactivation process on the Kekerengu fault by Holden et al.[4] However, our model suggests that the first peak stems from the earlier rupture of the Papatea segment (see Supplementary

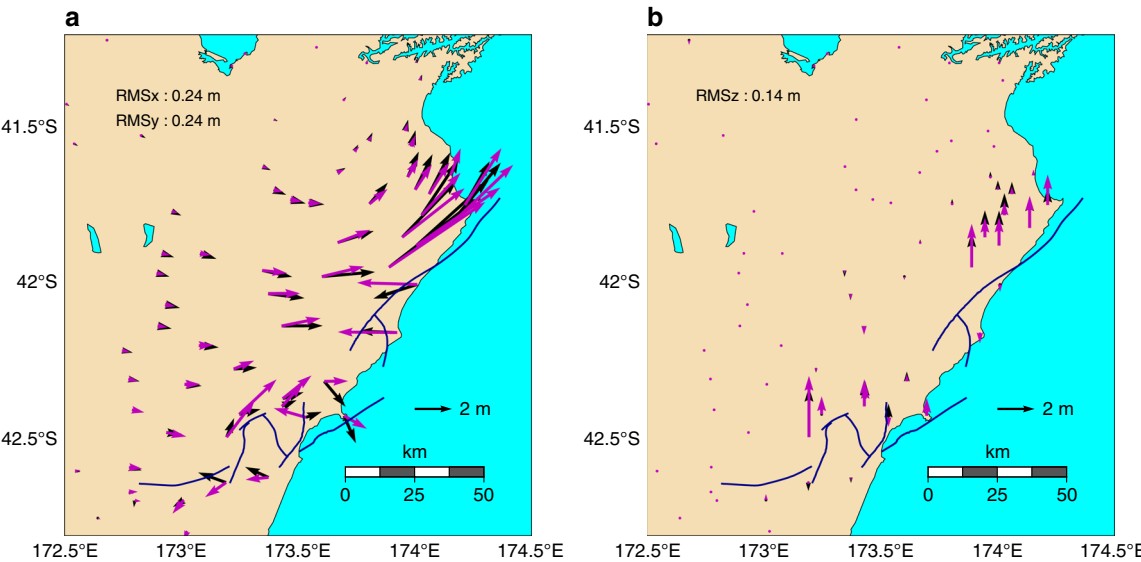

**Fig. 7** Comparison of observed and synthetic static ground deformation. Shown are observed (black) and modeled (magenta) horizontal (**a**) and vertical (**b**) ground displacement at Global Positioning System (GPS) stations. Root-mean-square (RMS) misfits are provided for each component. The observed ground displacements at the locations of the GPS stations are taken from Hamling et al.[1]

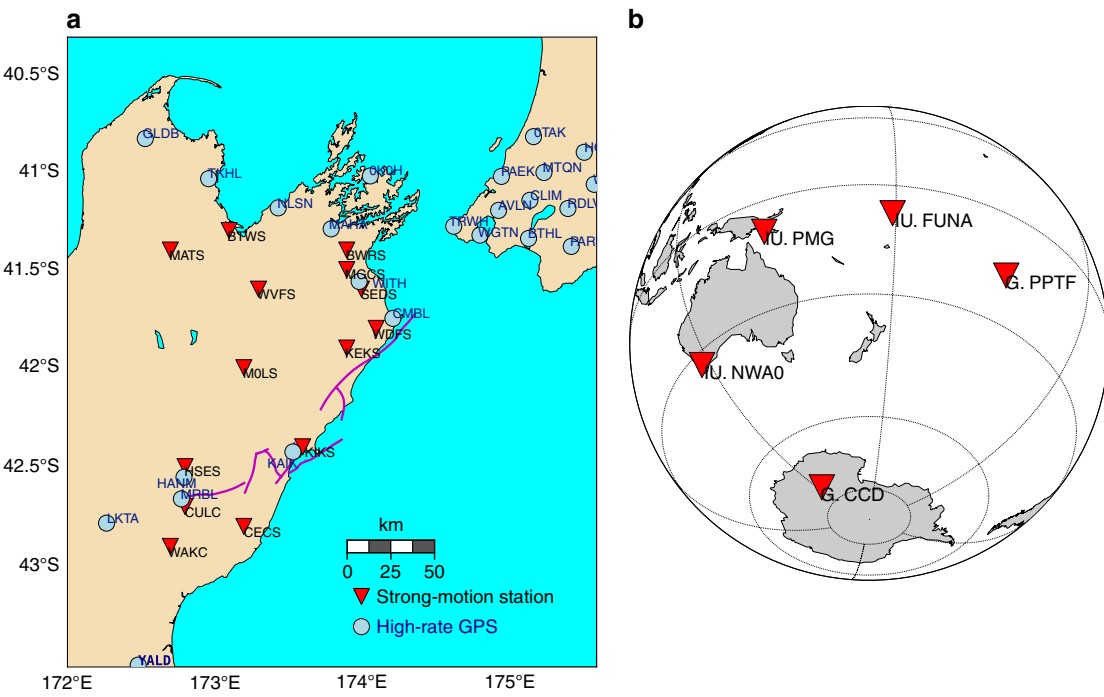

**Fig. 8** Locations of seismic and geodetic observations used for model verification. Near-fault high-rate Global Positioning System (GPS) and strong-motion stations (on South Island) actively recording during the Kaikōura earthquake (**a**). Teleseismic stations at which synthetic data is compared with observed records (**b**)

movie 2). The ground motions recorded at station KEKS are thus consistent with a sequential rupture from the Papatea to Kekerengu faults. Strong evidence for a rupture sequence from Papatea to Kekerengu is further provided by the teleseismic back-projection results of Xu et al.[7] More recently, comparing remote sensing and field observations to 2D dynamic simulation results, Klinger et al.[46] showed that observed patterns of surface slip and off-fault damage support this scenario.

**Teleseismic data.** Our model without slip on the subduction interface satisfactorily reproduces long-period teleseismic data.

Synthetics are generated at five teleseismic stations around the event (Fig. 8). We translate the dynamic fault slip time histories of our model into a subset of 40 double couple point sources. From these sources, broadband seismograms are calculated from a Green's function database using Instaseis[47] and the PREM model for a maximum period of 2 s including anisotropic effects. In the long period range considered (100–450 s) the fit to observations is satisfying (Fig. 10). The effect of gravity, significant for surface waves at those periods, is not accounted for in the synthetics due to methodological limitations of Instaseis. In conjunction with our restriction to the 1D PREM model instead of incorporating

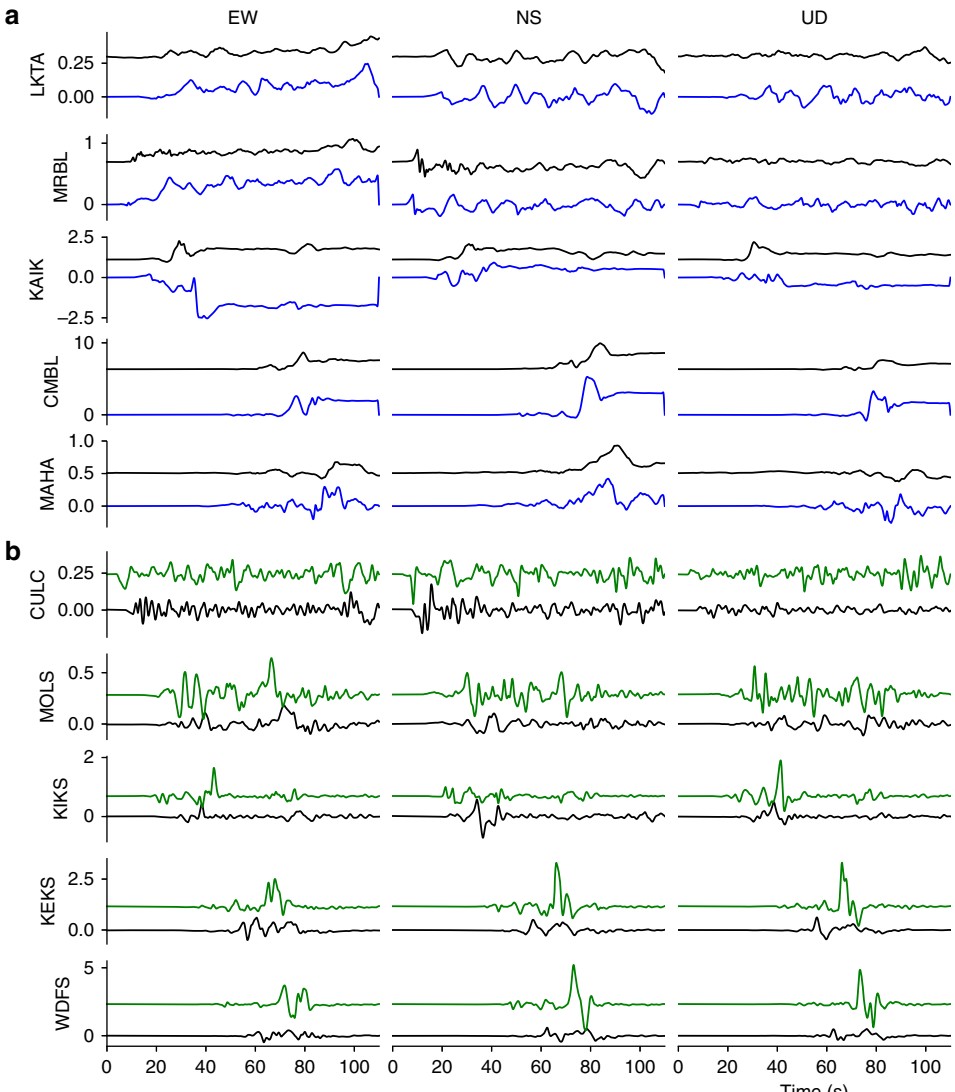

**Fig. 9** Comparison of modeled and observed ground motions. **a** Synthetic (blue) and observed (black) ground displacements at selected Global Positioning System (GPS) stations. A 1 s low-pass filter has been applied to both signals. **b** Synthetic (green) and observed (black) ground velocities at selected strong-motion stations. A 0.005–1 s band-pass filter has been applied to both signals. The station locations are shown in Fig. 8. The station names are labeled on the vertical axis of each plot

3D subsurface information, remaining differences between synthetics and observed records are expected. Following the same approach but based on Duputel and Riviera's[2] kinematic source model inferred from teleseismic data indeed yields similar discrepancies. Overall, our results imply that slip on the subduction interface is not required to explain teleseismic observables.

**Uniqueness of the dynamic model**. There is a high level of uniqueness in the outcome of our dynamic model. Slight variations on the initial conditions, for instance, a subtle change in the maximum principal stress direction of 10° degrees or a less transpressional regime (e.g., a 10% increase of the stress shape ratio defined in Eq. 9 in "Initial stresses" in the Methods section), lead to early spontaneous rupture arrest. Changes in fault geometry (orientation, size, and separation distance of fault segments) also affect the dynamics considerably. Moreover, ad hoc abrupt lateral changes in initial fault stress or strength are not required to steer the rupture along its zigzagged path. We nevertheless acknowledge the possibility of alternative models yielding similar rupture dynamics. Such models can be readily

designed based on the trade-offs we define in "Initial stresses" in the Methods section, for example, by decreasing or increasing the effects of deep stress concentrations, fluid pressure, or frictional weakening. In "Apparent fault weakness" in the Methods section, we accordingly ensure the robustness of important modeling choices of the preferred model.

**Linking fault segments**. Two segments, the Stone Jug and the Point Kean faults, are crucial for the successful propagation of the rupture to the North. The Stone Jug fault hosts little slip but allows the earthquake to branch towards the Hundalee fault. The offshore Point Kean fault links at depth the seemingly disconnected Southern and Northern parts of the fault system (as proposed by Clark et al.[10]), whose surface traces are separated by a large gap of 15 km. Our model matches the observed (horizontal) surface rupture in the Northern part[48], the inferred slip amplitude and the northwards rupture propagation on the Point Kean fault, by dominantly oblique faulting. It supports a previous suggestion that rupture of the Point Kean fault was responsible for the observed on-shore coastal uplift extending 20 km north of

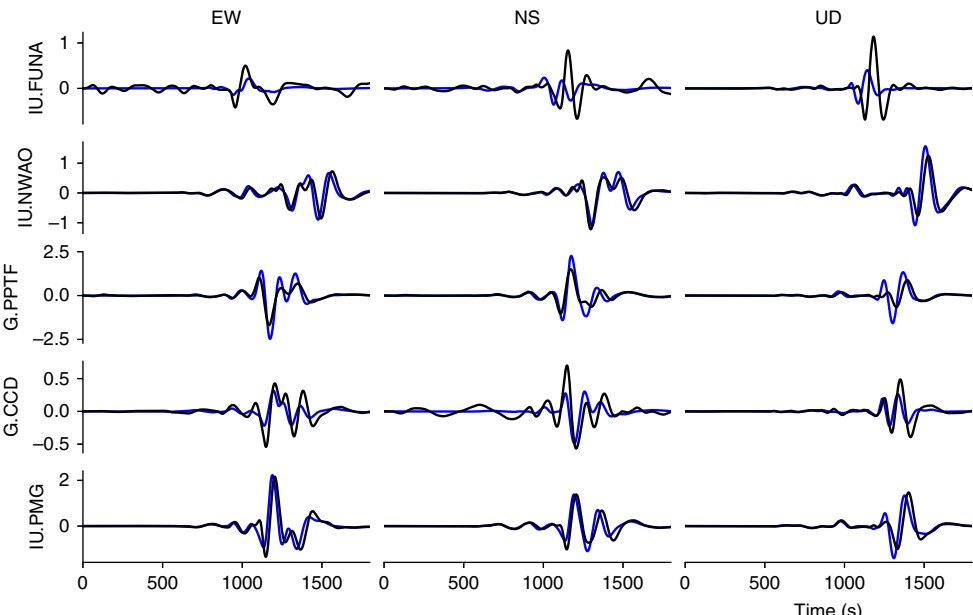

**Fig. 10** Comparison of modeled (blue) and observed (black) teleseismic waveforms. A 100–450 s band-pass filter is applied to all traces. Synthetics are generated using Instaseis[47] and the PREM model including anisotropic effects and a maximum period of 2 s

Kaikōura Peninsula[10]. On the other hand, a stronger dip-slip component would be required to explain the northeastward GPS displacements around this thrust fault. According to the dynamic rupture model, this could only be achieved by an (unlikely) local prestress rotation of about 30° towards South, or by considering a fault geometry with lower strike.

**Rupture complexity**. The dynamic model shows rupture complexity also at a fine scale. Rupture takes the form of slip pulses (Fig. 4) of various origins: fast-velocity-weakening friction promotes self-healing slip pulses[14,49], which can propagate at lower background stress levels and with smaller slip than crack-like ruptures. The nonlinear interaction between frictional failure and the free surface causes interface waves that bounce back from the surface, and fault ends; branching points lead to rupture front segmentation, and unloading stresses carried by seismic waves reflected from subsurface impedance contrasts cause healing fronts. The Hundalee, Point Kean, Papatea, and Kekerengu segments slip more than once.

Rupture complexity can affect seismological inferences of fault friction properties. Frictional parameters are typically adopted from laboratory experiments. However, it is uncertain how valid it is to extrapolate results from the laboratory scale to the field scale. For the Kaikōura earthquake, a large slip-weakening distance $D_c$, the amount of slip over which frictional weakening occurs, has been estimated from a strong-motion record[50]. Despite the much smaller on-fault $D_c$ values (0.2– 0.5 m) in our model, the apparent $D_c$ value inferred from the resulting off-fault ground motions is large (5.6 m, Supplementary Fig. 4), which can be attributed to intertwined waveforms from multiple slip fronts.

**Discussion**

The physics-based dynamic source modeling approach in this study has distinct contributions compared to the data-driven kinematic source modeling approach. In the latter, a large number of free parameters enables close fitting of observations at the expense of mechanical consistency. Furthermore, the kinematic earthquake source inversion problem is inherently non-unique (many solutions fit the data equally well). In contrast, our

dynamic model is controlled by a few independent parameters. Its main goal is to understand the underlying physics of the cascading rupture sequence. Adopting fault geometries and a regional stress state consistent with previous studies, our dynamic rupture model reproduces major observations of the real event, reveals unexpected features, and constrains competing hypotheses.

Our results provide insight on the state of stress under which complex fault systems operate. In our model, strong frictional weakening, fluid overpressure, and deep stress concentrations result in a remarkably low apparent friction. Yet, the low average ratio of initial shear stress to normal stress does not hinder dynamic rupture cascading across multiple fault segments. Instead, it is crucial to achieve the full cascading rupture with realistic stress drop and slip. In "Apparent fault weakness" in the Methods section, we discuss fault strength based on the orientation of the fault system, apparent fault strengths in the static and dynamic sense, and explore additional model setups demonstrating the robustness of our preferred model. We conclude by quantifying the relative contributions of our modeling assumptions to the apparent weakness of faults.

The effects of overpressurized fault fluids and deep stress concentrations and the additional effect of a low dynamic friction result in an overall low apparent friction coefficient. We find that reproducing all aspects of the rupture cascade requires all three effects. The combined effect of strong frictional weakening, fluid overpressure, and deep stress concentrations and the fundamental impact of fault weakness on the existence of subduction and tectonics (e.g. ref. [23]) show the importance of mechanical feedbacks across multiple time scales, from the short-term processes of dynamic rupture and earthquake cycles to the long-term geodynamic processes that shape and reshape the Earth.

Dramatic frictional weakening is one of the key mechanisms contributing to fault weakness in our model. Our assumed dynamic friction coefficient, $f_w = 0.1$, falls within the range of values typically observed in laboratory experiments and considered by the dynamic rupture community (e.g. refs. [14,15,25]). Nevertheless, we probed the necessity of such a low value by additional simulations, as detailed in Methods section "Alternative model setups". By static considerations, we find that a

sustained cascading rupture under a higher $f_w$ would require conditions that disagree with stress inversion inferences, namely a too low stress shape ratio (Supplementary Fig. 5). In addition, prescribing higher $f_w$ results in a prestress distribution of larger variability, less favorable for rupture cascading.

Frictional failure in our model initiates at the best-oriented fault segment, in contrast with the "keystone fault" model[51] in which large multi-fault earthquakes nucleate on a misoriented fault. The dynamic rupture cascade does not require laterally heterogeneous initial stresses, as those arising on fault networks in which optimally oriented faults release stress not only during large earthquakes but also via smaller events or aseismic creep.

We find that a zigzagged propagation path, accompanied by rupture delays at the transitions between faults, can explain apparently slow rupture speeds. While the surprisingly slow apparent rupture velocity and long rupture duration were depicted widely in seismological observations, our dynamic model provides a mechanically viable explanation for this observation.

Physics-based dynamic modeling contributes crucial arguments to the debate of whether the rupture of multiple crustal faults during the Kaikōura earthquake was promoted by slip on the underlying Hikurangi subduction interface. Rupture of the subduction interface is not favored by the regional stresses we inferred. A planar, shallowly dipping subduction interface approximated similar to previous studies[1] experiences very low shear stresses when included in our model. Dynamic triggering of such a subduction interface is further impeded by its large depth below the crustal fault network. However, slip may be promoted if the stresses rotate at depth or if the megathrust is frictionally weaker than the crustal faults (e.g. refs. [32,35]). We show that incorporating the shallowly dipping (35°) Point Kean fault segment successfully links the Southern and Northern parts of the fault system without involvement of the Hikurangi subduction interface. Our model is equally compatible with long-period teleseismic data as models assuming slip on the subduction interface and may be further tested by tsunami observations.

Features of the Kaikōura earthquake that remain unexplained by our dynamic models suggest opportunities to better understand the role of fault heterogeneities. These features include the inferred localized slip at depth on the Upper Kowhai fault as well as incompletely modeled aspects of the observed waveforms. Our dynamic rupture scenario is able to explain the early rupture termination to the South, but does not give a definitive answer concerning the origin of rupture termination to the North. On the Humps fault zone, spontaneous rupture termination to the West is observed, associated with a slight change in the strike direction resulting in a less favorable fault orientation. In additional dynamic rupture simulations including the nearest identified faults to the South, the Leonard Mound, and the Culverden reverse faults (e.g. ref. [52]), we found that the rupture is not able to trigger those faults. To the North, we have to locally reduce the initial stresses on the Northernmost part of the Needles fault to prevent its rupture with large slip. The very straight surface rupture of the Needles fault[48] does not suggest a high segmentation that may have prevented the rupture to extend further North. Hamling et al.[1] and Wang et al.[5] suggest a steeper geometry for this segment (dip angle of 70°), which would result in an increased shear over normal stress ratio, favoring rupture instead of terminating it. These considerations indicate that the most likely reason for the rupture termination to the North is the presence of an asperity to which the many aftershocks in the region[53] might be associated.

Our model provides a solution to one of the fundamental riddles of the Kaikōura earthquake: why did the rupture by-pass the Hope fault? The lack of significant slip observed on the Hope fault is surprising given its orientation similar to the Kekerengu fault, its fast geologic slip-rate and short recurrence interval (180–310 years[54] and references therein), and its linkage to most mapped faults involved in the rupture. In our model, the Hope and Conway-Charwell faults are <1 km apart at the surface, and diverge at depth because of their different dipping angles. Both faults are well oriented relative to the background stress. Yet, the Hope fault is not triggered by the rupture of the Conway-Charwell fault, nor later on by the rupture of the Hundalee and Point Kean faults. We interpret this non-rupture as a consequence of the restraining step-over configuration formed by the Conway-Charwell and Hope faults, leading to an unfavorable distribution of dynamic stresses on the Hope fault (e.g. ref. [55]). Dynamic modeling allows assessing the possibility of rupture jumping across such unconventional step-over configurations, combining thrust and strike-slip faulting mechanisms and faults of different dip angles.

Dynamic rupture modeling is now approaching a state of maturity and computational efficiency that should soon allow it to be integrated synergistically with data inversion efforts within the first days following the occurrence of an earthquake, making physics-based interpretations an important part of the rapid earthquake response toolset.

## Methods

**Numerical method.** We solve the coupled dynamic rupture and wave propagation problem using the freely available software SeisSol[56,57] (https://github.com/SeisSol/SeisSol) based on the Arbitrary high-order accurate DERivative Discontinuous Galerkin method (ADER-DG). SeisSol employs fully adaptive, unstructured tetrahedral meshes to combine geometrically complex 3D geological structures, nonlinear rheologies, and high-order accurate propagation of seismic waves. Our model (Fig. 3) includes a geometrically complex fault network, high-resolution topography[58], 3D subsurface structure[59], and plastic energy dissipation off the fault[60,61]. A high-resolution model is crucial for accurately resolving rupture branching and (re-)nucleation processes. The degree of realism and accuracy achieved in this study is enabled by recent computational optimizations targeting strong scalability on many-core CPUs[62–64] and a ten-fold speedup owing to an efficient local time-stepping algorithm[37]. Simulating 90 s on a computational mesh consisting of 29 million elements required typically 2 h on 3000 Sandy Bridge cores of the supercomputer SuperMuc (Leibniz Supercomputing Center, Germany), which is well within the scope of resources available to typical users of supercomputing centers. The few dynamic rupture simulations required to constrain the initial stress setup (see "Initial stresses" in Methods section) employed a coarser discretization of wave propagation in the volume while still finely resolving the faults, reducing computational cost by 80%.

**Mesh.** The domain is discretized into an unstructured computational mesh of 29 million high-order (spatio-temporal order 4) four-node linear tetrahedral elements (Fig. 3). The mesh resolution is refined to element edge lengths of 300 m close to faults. Topography and bathymetry are discretized by at most 1000 m and refined in regions of strong variations. The mesh allows resolving the seismic wavefield at frequencies up to 3 Hz in the vicinity of the faults.

**Fault friction.** We use a rate- and state-dependent friction law with fast velocity weakening at high speed proposed in the community benchmark problem TPV104 of the Southern California Earthquake Center[65] and similar to the friction law introduced by Dunham et al[28]. Here we provide the governing equations using the notations defined in Supplementary Table 1. The magnitude of the shear traction $\tau$ is assumed to always equal the fault strength, defined as the product of the friction coefficient $f$ and the effective normal stress $\sigma_n'$:

$$\tau = f(V, \psi)\sigma_n'. \tag{1}$$

The traction $\boldsymbol{\tau}$ and slip rate $\mathbf{V}$ vectors are parallel and satisfy:

$$\tau \mathbf{V} = V \boldsymbol{\tau}. \tag{2}$$

The friction coefficient $f$ depends on the slip rate $V$ and a state variable $\psi$:

$$f(V, \psi) = a \operatorname{arcsinh}\left(\frac{V}{2V_0} \exp\left(\frac{\psi}{a}\right)\right). \tag{3}$$

The state variable $\psi$ evolves according to the following differential equation:

$$\frac{\mathrm{d}\psi}{\mathrm{d}t} = -\frac{V}{L}\left(\psi - \psi_{ss}(V)\right), \tag{4}$$

where $\psi_{ss}$ is the value of the state variable at steady-state given by:

$$\psi_{ss}(V) = a \ln \left( \frac{2V_0}{V} \sinh \left( \frac{f_{ss}(V)}{a} \right) \right), \tag{5}$$

where the steady-state friction coefficient is

$$f_{ss}(V) = f_w + \frac{f_{LV}(V) - f_w}{\left( 1 + (V/V_w)^8 \right)^{1/8}} \tag{6}$$

and the low-velocity steady-state friction coefficient $f_{LV}$ is given by:

$$f_{LV}(V) = f_0 - (b-a)\ln(V/V_0). \tag{7}$$

At slip rates higher than the characteristic slip rate $V_w$, $f_{ss}$ asymptotically approaches the fully weakened friction coefficient $f_w$, with a decay roughly proportional to $1/V$. This feature of friction is observed in laboratory experiments and is present in thermal weakening theories. At low slip velocities, this friction law is consistent with classical rate-and-state friction.

The initial distribution of the state variable $\psi_{ini}$ is obtained, from Eqs. 1 and 3, assuming that the faults are initially at steady state, sliding at a slip rate of magnitude $V_{ini} = 10^{-16}$ m/s:

$$\psi_{ini} = a \ln \left( \frac{2V_0}{V_{ini}} \sinh \left( \frac{\tau_{ini}}{a\sigma_{ini}} \right) \right), \tag{8}$$

where $\tau_{ini}$ and $\sigma_{ini}$ are the (spatially varying) initial shear and normal tractions on the fault.

The values of the frictional properties adopted in this study are given in Supplementary Table 1. Some parameters are depth dependent, as indicated in Supplementary Fig. 6. To suppress shallow supershear transition, $V_w$ is assumed to be larger at shallow depth (e.g., ref. [15]) on all faults (except for the Leader segment, to avoid suppressing its emerging shallow rupture quickly after branching from the Humps fault).

We infer the equivalent slip-weakening distance $D_{c,eq}$ of our simulations from the resulting curves of shear stress as a function of slip at various points along the rupture. We define

$$D_{c,eq} = 2G_c / \left( \tau_{peak} - \tau_{final} \right) \quad \text{where } G_c = \int_{D_{peak}}^{\infty} (\tau(D) - \tau_{final}) dD. $$

Supplementary Fig. 7 shows the typical stress change at five fault locations. The values of $D_{c,eq}$ fall in the range from 0.2 to 0.5 m. In addition, following Kaneko et al.[50], we apply the method of Mikumo et al.[66] to our modeled seismograms at station KEKS (Supplementary Fig. 4) to estimate an apparent slip-weakening distance $D_c''$ defined as twice the fault-parallel displacement at the time the peak fault-parallel velocity is reached. The fault-parallel velocity waveform has two peaks of similar amplitude, separated by a few seconds, which may result from multiple slip fronts on the Kekerengu fault (see Supplementary movie 2). We estimate $D_c'' = 2.4$ m from the first peak. The second peak gives $D_c'' = 8.9$ m, larger than the value of 4.9 m estimated by Kaneko et al[50]. These $D_c''$ estimates are larger than the on-fault $D_{c,eq}$ for at least three reasons. First, the station is at a distance from the fault (~2.7 km) much larger than the maximum distance for resolution of Mikumo et al.'s[66] method. ($R_c = 0.8 V_s T_c = 23$ m, where $V_s = 2.9$ km/s is the shear wave velocity and $T_c = 0.1$ s is the breakdown time. Note that $T_c$ in our simulations is much smaller than the apparent value of 5.5 s reported by Kaneko et al.[50]) Second, off-fault plasticity (included in our model) can contribute to increase the apparent $D_c''$. Third, our dynamic model features multiple slip fronts contributing to the cumulative fault-parallel displacement, thus increasing $D_c''$.

**Off-fault plasticity.** We model off-fault dissipation assuming a Drucker–Prager elasto-viscoplastic rheology[61]. The failure criterion is parameterized by two material properties, internal friction coefficient and cohesion. We set the internal friction coefficient equal to the reference fault friction coefficient (0.6). Following Rotten et al.[67], we consider an empirically motivated depth-dependent distribution of cohesion (Supplementary Fig. 8) to account for the tightening of the rock structure with depth. Lower cohesion in the upper 6 km allows suppressing the unrealistic occurrence of shallow supershear transitions without preventing rupture cascading by dynamic triggering. A viscoplastic relaxation mechanism is adopted to ensure convergence of the simulation results upon mesh refinement. Its relaxation time $T_v$ also controls the effectiveness of plasticity. We set $T_v = 0.05$ s, independently of the mesh resolution. We consider depth-dependent off-fault initial stresses consistent with the initial stresses prescribed on the fault.

**Initial stresses.** Following Townend et al.[33], we first constrain the initial stress tensor using the parameters $SH_{max}$, $\nu$, and $\theta$. Following Lund and Townend[68], $SH_{max}$ is defined as the azimuth of the maximum horizontal compressive stress. It coincides with the commonly used horizontal projection of the largest sub-horizontal stress if the state of stress is Andersonian, that is, if one principal stress component is vertical. The stress shape ratio is defined as:

$$\nu = (s_2 - s_3)/(s_1 - s_3), \tag{9}$$

where $s_k$ are the amplitudes of the principal stresses. The angle $\theta$ is the orientation of the intermediate principal stress relative to the horizontal plane.

We set the initial stresses in the rupture area to be consistent with regional stress parameters inferred from earthquake focal mechanisms by Townend et al.[33], and their uncertainties. Among the earthquake clusters they considered, the ones within our region of interest are, from North to South, clusters 27, 65, 16, 11, and 18 (Supplementary Fig. 5-a). We ignore cluster 53, located between 50 and 100 km depth, because it is much deeper than the Kaikōura earthquake source. The stress parameters at the considered clusters are shown in Supplementary Fig. 5-b. The average $SH_{max}$ is 96° (the average over the whole South Island is 115°). The value of $\nu$ is inferred to be in the range of 0.4–0.5, but lower values cannot be ruled out. Note that we use a different definition of $\nu$ than Townend et al[33]. The value of $\theta$ falls in the range 80°–110°.

An additional parameter, the relative prestress ratio $R$ between fault stress drop and breakdown strength drop, allows constraining the magnitude of the deviatoric stresses:

$$R = \frac{\tau - \mu_d \sigma_n}{(\mu_s - \mu_d)\,\sigma_n}. \tag{10}$$

To compute $R$ we assume $\mu_d = f_w = 0.1$, as we observe that the fully weakened friction $f_w$ is typically reached in our simulations. The maximum friction coefficient reached during rupture ($\mu_s$) is not a prescribed model parameter. Its value varies along the fault and often exceeds $f_0$, but rarely falls below this value. For simplicity, we use $\mu_s = f_0 = 0.6$ as a conservative value: in our simulation results, the real $R$ can be smaller than the one we prescribe, but is rarely larger.

Following the notations of Aochi and Madariaga[34], we define

$$P = (s_1 + s_3)/2 \text{ and } ds = (s_1 - s_3)/2. \tag{11}$$

$(P, 0)$ is the center of the Mohr–Coulomb circle and ds is its radius. The $s_i$ are related to $P$, ds, and $\nu$ by:

$$\begin{aligned} s_1 &= P + ds, \\ s_2 &= P - ds + 2\,\nu\,ds, \\ s_3 &= P - ds. \end{aligned} \tag{12}$$

The effective confining stress $\sigma_c' = (s_1 + s_2 + s_3)/3$ is related to $P$ by:

$$\sigma_c' = P + (2\nu - 1)\,ds/3. \tag{13}$$

We assume a lithostatic confining stress given by $\sigma_c(z) = \rho g z$ and a rock density of $\rho = 2670$ kg/m³. In a transpressional regime, this results in an average stress $\sigma_c(z) = (s_1 + s_2 + s_3)/3 = \rho g z$, which is slightly lower than when adopting the conventional assumption of $\sigma_{zz}(z) = \rho g z$. Switching the depth dependence of stress while not altering stress drop and rupture dynamics in our model can readily be achieved by slightly adjusting fluid pressure.

We assume fluid pressure throughout the crust is proportional to the lithostatic stress:

$P_f = \gamma \sigma_c(z)$, where $\gamma$ is the fluid–pressure ratio. The effective confining stress is thus $\sigma_c'(z) = (1-\gamma)\sigma_c(z)$. The value $\gamma = \rho_{water}/\rho = 0.37$ corresponds to a hydrostatic state; higher values $\gamma > 0.37$ correspond to overpressurized states.

The shear and normal stresses $\tau$ and $\sigma_n$ on a fault plane oriented at an angle $\phi$ relative to the maximum principal stress are:

$$\begin{aligned} \tau &= ds \sin(2\phi), \\ \sigma_n &= P - ds \cos(2\phi). \end{aligned} \tag{14}$$

An optimally oriented fault plane is one that, under homogeneous initial stress and stressing rate, would reach failure before any other fault with different orientation. At failure, its shear to normal stress ratio is maximized (compared to other fault orientations) and equal to $\mu_s$. Its angle is:

$$\phi = \pi/4 - 0.5 \operatorname{atan}(\mu_s). \tag{15}$$

We will prescribe $R_{opt}(z) = R_0 g(z)$ on the (virtual) optimally oriented fault plane, where $g(z)$, described hereafter, is a stress modulation function accounting for stress concentrations expected right above the seismogenic depth of faults loaded by deep fault creep (Supplementary Fig. 2). Using Eqs. 10, 13, and 14, we solve for ds and obtain:

$$ds = \frac{\sigma_c'}{\sin(2\Phi)/\left( \mu_d + (\mu_s - \mu_d)R_{opt} \right) + (2\nu - 1)/3 + \cos(2\Phi)}. \tag{16}$$

For given values of $\nu$ and $R_0$, we can compute the depth-dependent $s_i$ using Eqs. 12, 13, and 16. The orientations of the three principal stress components (assumed depth-independent) are determined by the angles $SH_{max}$ and $\theta$ and by the constraint that the faulting mechanism on the optimally oriented plane is strike-slip. This defines a depth-dependent stress tensor $(b_{ij})$. The final stress tensor $(s_{ij})$ is obtained by applying a second stress modulation function $\Omega(z)$, which smoothly cancels the deviatoric stresses below the seismogenic depth $z_{seis}$ (Supplementary Fig. 2):

$$s_{ij} = \Omega(z)b_{ij}(z) + (1 - \Omega(z))\sigma_c'(z)\delta_{ij}. \tag{17}$$

The initial stress model depends on four parameters constrained by observations ($SH_{max}$, $\theta$, $\nu$, and $z_{seis}$) and on three unknown parameters related to fluid pressure, background shear stress, and deep stress concentration ($\gamma$, $R_0$, and $g$

(0)). To determine the preferred values adopted in our final simulations, instead of running costly dynamic rupture simulations for each parameter set, we developed the following workflow, illustrated in Supplementary Fig. 1.

In a first step, we constrain $SH_{max}$, $\theta$, and $\nu$ to ensure compatibility of the stress with inferred fault geometry and slip rake. As a first assumption, we use a fluid–pressure ratio $\gamma = 0.75$ [37]. We set uniform stress modulation functions, $g(z) = 1$ and $\Omega(z) = 1$, and assume $R_{opt}(z) = R_0 = 0.7$ on the optimal plane. We expect this $R_0$ value to be high enough to allow a sustained rupture on faults of highly varying orientations and low enough to result in a reasonable stress drop. An order-of-magnitude estimate of stress drop is $R_0(1 - \gamma)\sigma_c(\mu_s - \mu_d)$, under the assumption $R_{opt}(z) = R_0$. We test different stress configurations, by varying $SH_{max}$ in the range 50°–120°, $\nu$ in the range 0–0.5, and $\theta$ in the range 70°–110°. For each value of the ($SH_{max}$, $\theta$, $\nu$) triplet, we do the following: compute the principal stress components using Eqs. 11–16; obtain the principal stress orientations from $SH_{max}$, $\theta$, and the additional constraint that the faulting mechanism of the optimal plane is strike-slip; compute and visualize the distribution of $R$ and of the shear traction orientation resolved on the fault system (Supplementary Fig. 9). We then select the stress configuration ($SH_{max}$, $\theta$, $\nu$) that maximizes $R$ all along the fault system, especially around rupture transition zones to enable triggering, and that optimizes the alignment between initial fault shear tractions and the slip directions inferred by Xu et al[7]. We rerun the procedure with a lower and a larger $R_0$ (0.5 and 0.9, respectively) to confirm that the conclusion obtained with $R_0 = 0.7$ still holds. In the next step of our stress setup, we will determine the preferred value of $R_0$ based on dynamic considerations.

Supplementary Fig. 9 presents a few of the many cases we tested. Eight examples are shown, which correspond to all permutations of the following values: $SH_{max} = 100°$ and 115°, $\theta = 80°$ and 90°, and $\nu = 0.5$ and 0.15. The value $\nu = 0.5$ results in a favorable stress orientation only for the eastern part of the Humps Fault Zone and on the Conway-Charwell fault. Lower values of $\nu$ are required to obtain a favorable stress orientation on the other faults. Our preferred value is $\nu = 0.15$. The value $SH_{max} = 100°$ achieves the best overall alignment between initial shear tractions and target slip on all faults. We find that the angle $\theta$ has a limited influence within the range tested, and thus opt for the simplest assumption of an Andersonian stress regime: $\theta = 90°$.

In a second step, we constrain $\gamma$, $R_0$, and the shape of the initial stress modulation functions, $g(z)$ and $\Omega(z)$, to allow the rupture to cascade along the whole fault system with a realistic amount of fault slip. This is done by trial-and-error based on dynamic rupture simulations. To save computational resources, we do the trial simulations on a coarser mesh (except near the fault) and first only simulate the initial 25 s to test if rupture can be sustained on the highly segmented southern part of our fault structure. Our stress modulation function is described by a minimum number of parameters (the width of the stress concentration area, the seismogenic depth $z_{seis}$, and the stress concentration shape, described hereafter). It is designed to capture the essential features of the stresses caused by deep creep: it is peaked at the base of the seismogenic zone ($g(z) = 1$) and decays to $g(0) < 1$ at shallower depth to represent the background stress. Most probably, any function with these general features could be used to achieve similar dynamic rupture results. We define $z_{seis}$ as the depth at which $\Omega(z)$ starts to decrease. We set it equal to the average maximum depth of the slip patches inferred by Xu et al[7]. We define the width of the stress concentration area as the depth range above $z_{seis}$ in which $g(z) = 1$. We prescribe its value just large enough to enable rupture transfer driven by stress concentration. We find that the values $R_0 = 0.85$, $g(0) = 0.6$, and $\gamma = 0.66$ ensure a subshear rupture and slip amounts consistent with results of previous source inversion studies. Supplementary Fig. 2 depicts the resulting shape of the initial stress modulation functions $g(z)$ and $\Omega(z)$. We also consider a small lateral variation in the regional stress, summarized in Fig. 2 and described in the main text.

The strength of the stress concentrations in our model (through parameters $R_0$ and $g(0)$) is partially constrained by observed rupture properties. The average stress drop in a dynamic model affects the average fault slip, rupture speed, and rupture size, and is roughly

$$d\tau \sim R_0 g(0)\left(\mu_s - \mu_d\right)(1 - \gamma)\sigma_c. \tag{18}$$

A high average stress drop leads to supershear rupture and unrealistically large slip, whereas a low value results in rupture terminating too early. Equation 18 allows identifying trade-offs between modeling parameters. For instance, a high $g(0)$ can be compensated by an increased pore pressure $\gamma$. Some trade-offs of modeling parameters can nevertheless be mitigated by physical constraints. For instance, a too small value of $g(0)$ would lead to a stress drop too peaked in the deeper portion of the rupture (too marked stress concentration), which would be inconsistent with slip models from source inversion. Nevertheless, resolving the detailed shape of such stress concentration might be challenging because finite source inversion and interseismic geodetic studies suffer from poorer resolution at depth and entail smoothing due to regularization. In future work, the depth-dependency of stress could be constrained by seismic cycle modeling capable of handling complex fault geometries.

To probe the importance of deep stress concentration, we performed a new model DR1 comparable to our preferred model but omitting deep stress concentrations. We decrease $R_0$ and simultaneously adjust the fluid pressure ratio $\gamma$ to preserve the average stress drop, and find the smallest $R_0$ enabling the full rupture cascade. The model DR1 has $R_0 = R_{opt}(z) = 0.7$ and $\gamma = 0.7$

(Supplementary Table 2). Its final fault slip is roughly similar to the slip of our preferred model. However, this alternative model has drawbacks compared to observations. In particular, it is less realistic in terms of timing. Its overall rupture duration is about 10 s shorter than our best scenario. This difference is mainly due to quicker shallow rupture transitions, such as the Humps-Leader branching, which are made easier by the increased prestress at shallow depth. Although this alternative model does not compare as well with observations as our preferred model, we cannot exclude the existence of an equally well-performing model featuring less pronounced stress concentrations.

**Apparent fault weakness**. Our preferred model is characterized by a low value of the initial shear to normal stress ratio over most of the seismogenic zone (Supplementary Fig. 3). Yet, most of the modeled faults are relatively well oriented with respect to the regional stress field. In the following, we describe the relation between our model assumptions and fault weakness, first in the static, then in the dynamic sense. We explore additional models, to assess the robustness of our preferred model, and quantify the effects contributing to the apparent fault weakness in our model.

**Apparent fault weakness in the classical sense**. In the classical sense, the fault system is considered strong since it is well-oriented relative to the regional stress.

In the classical Andersonian faulting theory, the strength of a fault is related to its orientation relative to the regional stress, in particular to the angle $\psi$ between the fault surface and the direction of maximum principal stress. This theory assumes that optimal faults are uniformly stressed at failure prior to an earthquake, with a ratio of shear to normal stress ($\tau/\sigma_n$) equal to the static friction $\mu_s$ everywhere along the fault. Their angle $\psi$ is the optimal angle $\phi$ defined in Eq. 15. A typical value is $\phi = 30°$ for $\mu_s = 0.6$. Faults away from the optimal orientation have a lower $\tau/\sigma_n$. Under these assumptions an active fault is weak (fails at low $\tau/\sigma_n$) if its orientation $\psi$ differs significantly from the optimal angle $\phi$.

According to this theory, most of our fault system is relatively well oriented relative to the regional stress. In fact, about 60% of the area of the fault system is oriented at angles ranging from 10° to 50° relative to the maximum principal stress (Supplementary Fig. 10). We point out that in a transpressional regime, these considerations may be less meaningful than under tectonic stresses resulting in unique faulting mechanisms.

**Static apparent fault weakness**. Statically, the model features a low ratio of fault shear to normal stress despite being well oriented.

In the framework of dynamic rupture modeling, faults can be stressed well below failure ($\tau/\sigma_n$ much lower than $\mu_s$) almost everywhere and yet break spontaneously. Only a small portion of the fault needs to reach failure to nucleate a rupture. In our model $\tau/\sigma_n$ is low over most of the rupture area (median value 0.09) and yet most faults are well oriented relative to the maximum compressive stress. Because the spatially averaged stress ratio $\tau/\sigma_n$ at the time of failure is a natural measure of the macroscopic fault strength, the faults in our model can be considered apparently weak, in a macroscopic sense, despite their local strength $\mu_s$ being high and their orientation being close to optimal.

The apparent strength ($\tau/\sigma_n$) of optimally oriented faults is related to our model parameters as follows. In dynamic rupture simulations, a relative fault strength is typically defined with respect to the frictional strength drop. This is quantified by the relative prestress ratio $R$ in our study (Eq. 10):

$$R = \frac{\tau - f_w\sigma_n(1 - \gamma)}{(f_0 - f_w)\sigma_n(1 - \gamma)}. \tag{19}$$

One of our input model parameters is $R_0$, the maximum value of $R$ within the deep stress concentrations on optimally oriented faults. The background value of $R$ governing the shallower fault areas is given as $R_0 g(0)$. A smooth transition from this background value to the deep stress concentration is prescribed by the stress modulation shape function $g(z)$ (Supplementary Fig. 2). The ratio of shear to normal stress on optimally oriented faults is then:

$$\frac{\tau}{\sigma_n} = (1 - \gamma)((f_0 - f_w)R_0 g(z) + f_w). \tag{20}$$

By varying the value of $R_0 g(z)$ between 0 and 1, we can prescribe any value of $\tau/\sigma_n$ between $(1 - \gamma)f_w$ and $(1 - \gamma)f_0$ independently of the fault orientation. Nevertheless, it is important to note that the portions of the fault experiencing deep stress concentration are characterized locally by a higher $\tau/\sigma_n$ ratio.

**Dynamic apparent fault weakness**. Dynamically, the modeled faults weaken dramatically at co-seismic slip rates while stress drops are limited by the interplay between elevated fluid pressure and deep stress concentration.

In our model, we assume strong dynamic weakening ($f_w = 0.1$). This is motivated by the dramatic friction decrease observed in laboratory experiments at co-seismic slip rates and by the theory of thermal weakening processes (as detailed in "Friction" in Results section and in "Fault friction" in Methods section). Furthermore, previous dynamic rupture studies utilizing fast velocity weakening with low values of $f_w$ successfully reproduced rupture complexities, such as rupture reactivation and pulse-like ruptures, without assuming small-scale (potentially

tuned) heterogeneities. In our model, adopting such friction law enables full cascading rupture and realistic amounts of slip, in contrast with simplified friction laws, as discussed in "Initial stresses" in the Results section.

Under this assumption, a low level of prestress is required to achieve a reasonable stress drop. To this end, we consider here two effects rarely taken into account together in dynamic rupture scenarios: (1) increased fluid pressure and (2) deep stress concentrations. We discussed their trade-offs in more detail in "Initial stresses" in Methods section. For example, very high values of fluid pressure alone could enable a suitable level of stress drop. However, model DR1 in "Initial stresses" in Methods section illustrates that the slow apparent rupture speed can only be reproduced by a model featuring stress concentrations at depth. We infer that the interplay of deep creep, elevated fluid pressure, and frictional dynamic weakening govern the apparent strength of faults and that these factors cannot be treated in isolation for such complex fault systems.

**Alternative model setups**. In our preferred model we assumed a fully weakened friction coefficient $f_w$ of 0.1. Here we present additional dynamic rupture experiments performed with higher values of $f_w$ as summarized in Supplementary table 2 to probe the robustness of our preferred model.

Increasing $f_w$ decreases the relative prestress ratio $R$ on most of the faults (Supplementary Fig. 11). To restore the rupture potential of these faults, the stress shape ratio (Eq. 9) must be decreased accordingly (Supplementary Fig. 12). The resulting values of $v$ are in stronger disagreement with stress inversion results than our preferred model with $f_w = 0.1$ and $v = 0.15$ (Supplementary Fig. 5b). Also, the resulting spatial distribution of prestress (under decreased $v$) has larger variability, which may hinder rupture cascading.

We performed two dynamic rupture simulations with increased $f_w = 0.3$ to probe the robustness of our assumption of low dynamic frictional resistance. In both models, $v$ is decreased from 0.15 to 0.05 to restore the rupture potential and the fluid pressure ratio $\gamma$ is decreased to retrieve a stress drop comparable to the one of the preferred model. The nucleation area is increased to account for the change in critical nucleation size. Both models differ only in their deep stress concentration. The first model, DR2, has similar deep stress concentration patterns as our best model. In the second model, DR3, we remove the depth dependence of the prestress ratio; that is, we set $R(z) = 0.85$ and $g(z) = 1$ above the stress tapering area, and we adjusted the fluid pressure ratio $\gamma$.

In model DR2 the rupture did not propagate successfully beyond the first rupture branching point connecting the Humps and Leader faults. It nevertheless yields a realistic amount of slip on the Humps fault zone, which confirms that the stress drop is unchanged. The second model, DR3, results in rupture branching towards the Leader fault but dies out at the next step-over, probably because of the now too low prestress on the shallow parts of the Southern Leader fault.

**Quantifying the relative contributions to apparent fault weakness**. The effects of overpressurized fault fluids and deep stress concentrations and the additional effect of a low dynamic friction result in a low apparent friction coefficient $\mu^*$ which can be approximated as:

$$\mu^* \sim \left(\mu_d + (\mu_s - \mu_d)g(0)R_0\right)(1 - \gamma). \tag{21}$$

Together with Eq. 20 this allows us to quantify the relative contribution of each effect to the fault apparent weakness in our preferred model: fluid overpressure $(1 - \gamma)$, deep stress concentration $(g(0)R_0)$, and dynamic weakening $(\mu_d)$. In our preferred model $\mu_d = 0.1$, $1 - \gamma = 0.33$, and $(\mu_s - \mu_d)g(0)R_0 = 0.26$.

Our unsuccessful attempt to reproduce all aspects of the rupture cascade in a model omitting stress concentrations (model DR1, Supplementary Table 2) illustrates that all three effects are important in allowing complex fault systems to operate at low apparent friction. Our findings warrant studies of the mechanical feedbacks between long-term geodynamic processes and the short-term processes of dynamic rupture and earthquake cycles.

**Rupture nucleation**. Rupture is nucleated by overstressing an area centered at the hypocenter, smoothly in space and time. This is achieved by increasing the initial relative prestress ratio $R_0$ as:

$$R_{0\,nuc} = R_0 + F(r)G(t). \tag{22}$$

$F(r)$ is a Gaussian-shaped function:

$$F(r) = 5\exp\left(\frac{r^2}{r^2 - r_c^2}\right) \quad \text{if } r < r_c, \\ = 0 \quad \text{elsewhere,} \tag{23}$$

where $r_c = 2$ km is the nucleation radius. The coefficient 5 is determined by trial-and-error numerical experiments to allow nucleation of sustained subshear rupture. $G(t)$ is a smoothed step function:

$$G(t) = \exp\left(\frac{(t-T)^2}{t(t-2T)}\right) \quad \text{if } 0 < t < T, \\ = 1 \quad \text{if } t \geq T, \tag{24}$$

where $T = 0.5$ s is the nucleation time.

## Code availability

We used the SeisSol (master branch, version tag 201807_Kaikoura) available on Github. The procedure to download, compile, and run the code is described in the code documentation (https://seissol.readthedocs.io).

## Data availability

The authors declare that all data supporting the findings of this study are available within the paper and its Methods section. In particular, all data required to run a simulation of the Kaikōura earthquake can be downloaded from https://zenodo.org/record/2538024. We provide a detailed readme file summarizing the data and data formats provided. We used the following projection: WGS 84/UTM Mercator 41 (EPSG:3994).

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

## Acknowledgements

The work presented in this paper was supported by the German Research Foundation (DFG) (projects no. KA 2281/4-1, GA 2465/2-1, GA 2465/3-1), by BaCaTec (project no. A4), by KONWIHR—the Bavarian Competence Network for Technical and Scientific High-Performance Computing (project NewWave), by the Volkswagen Foundation (ASCETE, grant no. 88479), by KAUST-CRG (GAST, grant no. ORS-2016-CRG5-3027 and FRAGEN, grant no. ORS-2017-CRG6 3389.02), by the European Union's Horizon 2020 research and innovation program (ExaHyPE, grant no. 671698 and ChEESE, grant no. 823844), by NSF CAREER award EAR-1151926, by the French government through the UCAJEDI Investments in the Future project ANR-15-IDEX-01 managed by the National Research Agency (ANR), by the Hong Kong Polytechnic University startup grant (1-ZE6R), and by the Hong

Kong Research Grants Council Early Career Scheme Fund (F-PP4B). Computing resources were provided by the Institute of Geophysics of LMU Munich[69], the Leibniz Super-computing Center (LRZ, projects no. h019z, pr63qo, and pr45fi on SuperMUC). We thank J. Townend for sharing his stress inversion data, J. Zhang and M. Vallée for sharing moment rate functions, C. Holden and E. d'Anastasio who provided processed GPS time-series, GNS Science for providing active fault database, earthquake rupture maps and reports, continuous GPS data, and strong-motion waveform data.

## Author contributions

This project was initiated by J.-P.A. Modeling was conducted by T.U. under the supervision of A.-A.G. with input from J.-P.A. and W.X. The manuscript was written jointly by T.U., A.-A.G., and J.-P.A.

## Additional information

**Competing interests:** The authors declare no competing interests.

