## [Peer Review File · Nature Communications]

Reviewers' comments:

Reviewer #1 (Remarks to the Author):

Review of the paper "Dynamic viability of the 2016 Mw 7.8 Kaikoura earthquake cascade on weak crustal faults" by T. Ulrich and coauthors.

This is a nice piece of work, providing new insights in the rupture evolution of the Kaikoura earthquake from a dynamic modeling perspective. Beside the existing number of publications on this outstanding earthquake, this work provides new, important findings, which deserve a prompt publication.

The paper provides a new dynamic model, able to explain the complex cascade rupture on multiple faults and slow apparent rupture duration, providing novel and convincing models and results. This is for sure an important publication, at least for a broad seismological community.

Thus I suggest acceptance upon minor revision. I list my few minor comments below.

1) Fault geometry

At Pag. 5, authors state that Southern part of Hope and the complete Upper Kowhai fault are removed from the model. While they justify this choice according to observation, it remains unclear what would have prevented these faults to fail.

2) Results

At Pag. 11, one of the main results is presented: the rupture cascade is dynamically viable without slip on the interface. This is the obvious result of the chosen model, which does not include the interface. Thus, I understand that results cannot disprove some slip on the interface. Some comments should probably be included here, or best in the discussion section.

3) Slow rupture speed

Again at Pag. 11, 3rd paragraph, is one of the most interesting results in my view. The rupture cascade contributes to explain the slow apparent rupture speed. I suggest the authors to stress more on this result, also in the discussion/conclusion sections. While the slow apparent velocity and long rupture durations have been depicted by almost all seismological works on Kaikoura, this is for me the first valuable justification for such observation, and strongly support the new dynamic model.

4) Slip-weakening distance D_c at pag. 13 should be better introduced

5) Typo:

Pag. 3 Ceska et al. → Cesca et al.

Reviewer #2 (Remarks to the Author):

Through a comprehensive, and well documented approach, the paper addresses an interesting question as to whether physics-based, dynamic rupture simulation can reproduce a set of observations associated with the complex Kaikōura (New Zealand) earthquake. The dynamic rupture model is based on a state-of-the-art numerical technique that includes realistic complexities, such as non-planar fault geometry, laboratory-derived friction laws, off-fault plasticity, material heterogeneities, and topography. This would be the first study to present dynamic rupture simulation of the complex Kaikōura earthquake. While findings about the rupture able to cascade without any contribution from the subduction interface are not able to rule the interface contribution totally out, they do bring an important conclusion from a dynamic perspective. However, I feel that three key issues need to be addressed to make this manuscript eligible for publication in Nature Communications: broader objectives, improved data fits to support the model and concerns about the model being overly-tuned. These three issues are described in more details below.

General comment: the authors have covered the method in great details in the supplementary section, sometimes leaving the main section too vague (e.g., p8 “using static considerations, we first aim for optimal stress parameters within their identified uncertainties”).

(I) Broader objectives

The study’s purpose is to unravel the “earthquake’s riddles in a physics based manner”, the 3 riddles being:

- a large gap separating fault segments according to Hamling et al., (2017) and Bradley et al. (2017),
- possibility of significant slip on the subduction interface
- and slow apparent rupture speed.

Regarding “the large gap”, it is worth noting that the Hamling et al. (2017) paper mentions “a large apparent gap”, as the fault are connected at depth in his model, and that the Bradley (2017) paper was a very preliminary attempt to obtain a kinematic model of the earthquake, however the waveform fits are not robustly supporting the suggested “20 sec” offset between ruptures.

While the questions are of scientific interest, I feel that the driving motivation in terms of making a significant scientific and societal impact using advanced physics methods was not only to explain why the earthquake ruptured in such ways but also most importantly why it did not rupture at all as expected. Three more riddles should be investigated about the Kaikōura earthquake:

- rupturing exclusively to the North even though the recent Canterbury earthquakes significantly increased stress south of the epicentre,
- bypassing the Hope Fault, despite it being the major regional source of seismic hazard,
- and finally rupturing 200 km of crustal faults but stopping short just south of Wellington.

A suggested approach could be to first validate the model of the earthquake rupture and appropriate physics parameters, then to test/discuss these against highly expected rupture scenarios which however did not occur during the Kaikōura sequence.

(II) Improved data fits

While I appreciate that dynamic rupture model performance is in validating overall rupture

processes, I have some concerns in the robustness of the results regarding quantitative comparison between modelled features and observed ones. Figure 5 coseismic displacement shows large discrepancies for the southernmost faults (EW and UD) and the Jordan and Papatea faults (opposite rupture mechanisms). Figure 6 GPS vector displacements also shows very large discrepancies with too much slip and inconsistent mechanism modelled to the south and too little slip to the north of the Kaikōura rupture. With regard to the proposed model time history evolution, Figure S4 synthetic waveforms for station KEKS shows over 10 second delay with actual recorded waveforms, therefore opening some concerns about the overall robustness of the proposed dynamic model. These points are described in more detail below.

+ While I understand that the dynamic rupture model is tuned to fit the source time function estimated from published kinematic models, removing the Upper Kowhai fault out of the fault model cannot be justified, as there is strong geodetic evidence suggesting that a moderate amount of slip at depths on Upper Kowhai occurred during the Kaikōura earthquake (Hamling et al., 2017; Xu et al., 2017). This type of consideration should be motivated by the data. Local strong-motion and high-rate GPS data (both available via GeoNet: <https://www.geonet.org.nz/>) do not really support the sequential rupture from Point Kean, to Papatea, to Kekerengu. In addition, Figure 6 shows a significant disagreement between the model and GPS data around the town of Kaikōura and the Upper Kowhai fault. By the way, please provide root-mean-square misfit or variance reduction, as the fits shown in Figure 6, especially the vertical component, appear to be not good. Also, please move Figure S1 to the main paper, which nicely highlights several key assumptions in this study.

+ Furthermore, published papers have not yet shown any evidence that the Papatea fault was ruptured before the Kekerengu fault. If anything, this is the other way around. Comparison of synthetic seismograms at station KEKS (Figure S4) with the data presented in published papers suggests a significant time shift (of more than 10 seconds) between the arrivals of the dominant phases at station KEKS, indicating that the model is not consistent with local seismic data. If the authors simply shift the rupture time of the Kekerengu fault to match the seismic data, then the source time function would become incompatible with those estimated from kinematic rupture models (Figure 4f). Please discuss these caveats and carefully interpret the modeling result.

+ The authors imply crustal faults ruptured during the Kaikōura earthquake are weak, citing evidence that crustal faults in the Marlborough region are not optimally oriented with respect to the regional stress inferred from seismological observations (Townend et al. 2012). Townend et al.'s inference was for major Marlborough faults (e.g., Hope, Clarence, Awatere, and Wairau) that are vertical strike-slip. In contrast, many faults (Hundalee, Jordan, Kekerengu, Needles, etc) ruptured during the Kaikōura quake dip toward northwest (50 – 70 degrees) and hence are more or less optimally oriented with respect to the regional stress field. Therefore I am puzzled by the authors' claim about the weakness of these faults. I would argue that the weakness of these crustal faults is simply the outcome of the assumed friction laws (not the outcome of the modeling result), in that in order to produce reasonable stress drop with enhance dynamic weakening, the prestress needs to be quite low (e.g., Noda et al., 2009).

(III) Model over-tuned

As a result of the complicated modeling approach, it is difficult to understand the robustness of the

claims made in this study and sensitivity of the results to underlying assumptions. My impression is that the model is overly tuned to match observations and kinematic rupture models.

+ The authors assume a particular shape of the depth-dependence stress functions shown in Figure S9, which are rather arbitrary aside from the fact that some increase of stress at depths near the rheological transition is intuitively expected. Why not simply assume a linear increase with depth? It looks to me that the exact shape of these functions strongly controls the rupture propagation behavior (Figure S1). Additional simulations and discussion would be needed to quantify the sensitivity of these assumed functions to the resulting rupture behavior.

+ On reducing initial stress on the Needles fault. If stress changes on the Needles fault induced by the 2013 Cook Strait sequences are used to justify the removal of 10 m coseismic slip, there is something wrong with the estimate of the stress changes. Note that the Cook Strait and Lake Grassmere earthquakes are M6.5-6.6, with the maximum slip of the order of 3 m, which would have induced relatively small stress changes (<0.5 MPa) on the Needles fault.

Reviewer #1 (Remarks to the Author):

R1_1) Fault geometry

At Pag. 5, authors state that Southern part of Hope and the complete Upper Kowhai fault are removed from the model. While they justify this choice according to observation, it remains unclear what would have prevented these faults to fail.

To address this important question, we ran additional simulations based on an updated fault geometry. Our new results elucidate the dynamic reasons for the non-rupture of the Hope fault. However, explaining the inferred moderate slip on the Upper Kowhai fault which is well oriented with respect to the regional stress seems to require fine-scale features beyond the scope of our modeling. Details are given below.

Updated fault geometry:

The fault geometry used in the submitted manuscript was constrained by observations accessible at the time of writing. This geometry is too simple to discuss the non-rupture of the Hope fault. In fact, in our original model, a small portion of the Hope fault ruptured and connected the rupture from the Conway-Charwell fault zone to the Stone Jug fault. However, surface rupture mapping (Litchfield et al. (2017), Nicol et al., (2018)) suggests that the Conway-Charwell fault, a few km South of the Hope fault, is likely to have enabled rupture transfer to the Stone Jug fault. Note that the fault named “Conway-Charwell fault” is different from the fault we termed “Conway-Charwell fault zone” in our original manuscript. The latter corresponds roughly to the “Leader fault” in detailed surface rupture maps (Litchfield et al. (2017), Nicol et al. (2018)).

We develop in our revised manuscript a new fault geometry model (fig R1), with a more complete account for fault complexity in the hypocentral region based on recently published data (Nicol et al. , 2018). The remainder of the fault geometry, outside the hypocentral region, is the same as in our original submitted model (figs. 1 and 2 of the revised manuscript). Specifically, the Conway-Charwell fault zone is now replaced by the Leader fault, which exhibits increasingly steeper orientation to the North (its dip is varying from 45° in its Southern end to 70° to the North), as suggested by Nicol et al. (2018). The small portion of the Hope fault considered rupturing in our original model is now replaced by the Conway-Charwell fault, which steps over the Leader fault and steepens slightly to the North (dip varying from 70° to 80°). Finally, we adjust the dip of the Stone Jug fault to the much

steeper dip angle of 80° South East. This value is closer to the range $85^\circ \pm 5^\circ$ North West inferred by Nicol et al. (2018). Our value allows a longer rupture path on the Hundalee fault and therefore an increased overall rupture duration, in better agreement with observation, than the geometry based on Nicol et al. (2018)'s dip inference.

Fig R1: The new fault geometry used in the revised manuscript (grey surfaces) compared in the hypocentral region with the fault geometry of the original model (transparent blue).

We considerably extend the paragraph in section 1.1 summarizing the observational constraints and our adaptations of the fault geometry. We merged in there most of the content of the previous appendix section A3. It now reads:

"We extend this simplified model to capture the complexity of the southern part of the fault network. The western tip of the Humps segment is slightly rotated (azimuth direction from WSW to W) in our model. The improved agreement with the mapped surface rupture enables spontaneous termination of the westward rupture front. We substitute the Conway-Charwell fault zone by the distinct Leader and Conway-Charwell faults (Nicol et al., 2018). The geometry of the Leader fault is similar to the Conway-Charwell fault zone of Xu et al. (2018)'s model, however the former is increasingly steeper to the North. Surface rupture mapping suggests a segmentation of the Leader fault in at least two segments (Nicol et al., 2018). Yet the continuity of the inferred ground-deformations in that region (Nicol et al., 2018) suggests a unified segment. Dynamic rupture experiments accounting for a large step-over within the Leader fault also suggest that a segmented geometry is not viable. The Conway-Charwell fault steps over the Leader fault. It runs roughly parallel to the Hope fault to the North. The Southernmost part of the long listric segment of Xu et al. (2018)'s geometry, representing the Hope fault, is replaced here by the Hope fault geometry proposed by Hamling et al. (2017), which is more consistent with the mapped fault trace and

inferred dip angle (Litchfield et al., 2014). The 60° dipping Stone Jug fault of Xu et al. (2018) is replaced by a steeper fault, as suggested by Nicol et al. (2018). The Hundalee segment is shortened at its extremities, to limit its slip extent according to Xu et al. (2018)'s inversion results.

Based on experimental dynamic rupture simulations, we remove the Upper Kowhai fault. Instead, we postulate that the previously unknown Point Kean fault (Clark et al., 2017) acted as a crucial link between the Hundalee fault and the Northern faults. The Upper Kowhai fault is well oriented relative to the regional stress and, when included, experiences considerable slip in contradiction with observations. Although geodetic data suggest a moderate amount of slip on this fault at depth (Hamling et al., 2017; Xu et al., 2018), we hypothesize that such slip is not crucial for the continuation of the main rupture process. This is supported by recent evidence suggesting the rupture propagated from the Papatea fault to the Jordan thrust (more details in sec. 2), rather than a Jordan thrust - Papatea fault sequence mediated by slip on the Upper Kowhai fault. Moreover, localized slip at depth on the Upper Kowhai fault would be difficult to reproduce without additional small scale features in the fault geometry or fault strength heterogeneities.”

We then updated slightly the initial stress following our proposed workflow (section A.6) and ran a new simulation. In comparison to the model of the original manuscript, the rupture properties are mostly unchanged (fig 4, 5f) and the agreement with observed ground-deformations in the hypocentral region is significantly improved (figs. 6 and 7).

Non-rupture of the Hope fault:

We updated the corresponding paragraph of the discussion section (sec. 3) in light of our new results with the updated fault geometry:

“Our model provides a solution to one of the fundamental riddles of the Kaikoura earthquake: why did the rupture by-pass the Hope fault? The lack of significant slip observed on the Hope fault is surprising given its orientation similar to the Kekerengu fault, its fast geologic slip-rate and short recurrence interval (180-310 years, Stirling et al. (2017) and references therein), and its linkage to most mapped faults involved in the rupture. In our model, the Hope and Conway-Charwell faults are less than 1 km apart at the surface, and diverge at depth because of their different dipping angles. Both faults are well oriented relative to the background stress. Yet, the Hope fault is not triggered by the rupture of the Conway-Charwell fault, nor later on by the rupture of the Hundalee and Point Kean faults. We interpret this non-rupture as a consequence of the restraining step-over configuration formed by the Conway-Charwell and Hope faults, leading to an unfavourable distribution of dynamic stresses on the Hope fault (e.g. Oglesby, 2005). Dynamic modeling allows assessing the possibility of rupture jumping across such unconventional stepover configurations, combining thrust and strike-slip faulting mechanisms and faults of different dip angles.”

We added a sentence in the first paragraph of the introduction summing up this achievement: “Our results associate the non-rupture of the Hope fault, one of the fundamental riddles of the event, with unfavourable dynamic stresses on the restraining step-over formed by the Conway-Charwell and Hope faults.”

Non-rupture of the Upper Kowhai fault:

A moderate amount of slip at depth on the Upper Kowhai fault is admissible from geodetic data (Wang et al., 2018, Xu et al., 2018). Hamling et al. (2017) suggest significant slip on the Upper Kowhai fault, with up to 15 m of slip in a narrow shallow band and 3 to 6 m of slip at depth. Such large slip may compensate in their model the absence of the Papatea fault, which generated significant ground deformations. In fact, models including the Papatea fault (e.g. Wang et al. (2018), Xu et al. (2018)) suggest much lower magnitude of slip. Fig R2 shows that most of the Upper Kowhai fault slipped less than 2 m according to Xu et al. (2018, model 3) (Note that the ticks labelling the thrust faults relatively to the main fault segment in Xu et al. (2018) figures 4 and 5 are wrongly located). Fault slip between 2 and 4 m is inferred only in a narrow area of the Upper Kowhai fault, in the vicinity of the Southern extremity of our Jordan Thrust model. We hypothesize that such slip is not crucial for the continuation of the main rupture process. This is supported by recent evidence suggesting a Papatea - Jordan thrust sequence (more details in sec. 2), contrary to a Jordan thrust - Papatea fault sequence supported by slip on the Upper Kowhai fault.

The Upper Kowhai fault is as well oriented relative to the regional stress as the Kekerengu fault. If we include the Upper Kowhai fault in our dynamic model, it breaks entirely with about 10 m of slip, which is too large compared to the observational inferences described above. An asperity may have prevented rupture on the shallowest part of the Upper Kowhai fault, leading to moderate slip at depth. We could have accounted for such asperity by removing only the shallow part of the Upper Kowhai fault, but we preferred to remove this whole fault to avoid over-tuning our model and to keep the geometry as simple as possible.

A paragraph summing up this point has been added to section 1.1 (see answer above to 'Updated fault geometry').

Fig R2: Fault geometry considered here (transparent blue) overlaid with Xu et al. (2018, model 3)'s fault slip, plotted with a saturated color scale to highlight the main areas of slip.

R1_2) Results

At Pag. 11, one of the main result is presented: the rupture cascade is dynamically viable without slip on the interface. This is the obvious result of the chosen model, which does not include the interface. Thus, I understand that results cannot disproof some slip on the interface. Some comments should probably be included here, or best in the discussion section.

As discussed in the third paragraph of the discussion section of the original manuscript, dynamic modeling contributes crucial arguments to the question of the involvement of the subduction interface in the Kaikoura earthquake. First, the subduction interface is not favorably oriented with respect to the regional stresses. As an illustration, prescribing a model-wide uniform $R_{op}^t(z) = 0.7$ on the optimal plane, as we did for compiling figure S7, leads to a low value $R = 0.08$ on the subduction interface oriented as in Hamling et al. (2017). Second, the subduction interface is at least 10 km apart from the slip area of our crustal fault network. A subduction interface rupture driving rupture on the crustal faults would require an exceptionally wide dynamic rupture jump (e.g. Wesnousky (2006), Bai and Ampuero (2017)).

Nevertheless, we acknowledge in sec.3 that rupture of the subduction interface cannot be excluded, and could have been promoted by tectonic stresses rotating at depth or a frictionally weak megathrust.

We added long-period teleseismic synthetics to the revised manuscript, which support our purely crustal faults model. The results section is extended by figs. 8 and 10 and the following paragraph:

“Our model without slip on the subduction interface satisfactorily reproduces long-period teleseismic data. Synthetics are generated at 5 teleseismic stations around the event (fig. 8). We translate the dynamic fault slip time histories of our model into a subset of 40 double couple point sources. From these sources, broadband seismograms are calculated from a Green’s function database using Instaseis (Krischer et al., 2017) and the PREM model for a maximum period of 10 s including anisotropic effects. In the long period range considered (100 to 450 s) the fit to observations is satisfying (fig. 10). The effect of gravity, significant for surface waves at those periods, is not accounted for in the synthetics due to methodological limitations of Instaseis. In conjunction with our restriction to the 1D PREM model instead of incorporating 3D subsurface information, remaining differences between synthetics and observed records are expected. Following the same approach but based on Duputel and Riviera (2017)’s kinematic source model inferred from teleseismic data, indeed yields similar discrepancies. Overall, our results imply that slip on the subduction interface is not required to explain teleseismic observables.”

R1_3) Slow rupture speed

Again at Pag. 11, 3rd paragraph, is one of the most interesting results in my view. The rupture cascade contributes to explain the slow apparent rupture speed. **I suggest the authors to stress more on this result, also in the discussion/conclusion sections.** While the slow apparent velocity and long rupture durations have been depicted by almost all seismological works on Kaikoura, this is for me the first valuable justification for such observation, and strongly support the new dynamic model.

We agree that this result is very interesting. In the results section we now illustrate how segmentation and the non-straight rupture path contribute to increase the rupture duration in the Southern region:

“Specifically, the modeled rupture sequence takes about 30 s to reach the Hundalee fault after nucleation, whereas a hypothetical, uninterrupted rupture propagating at a constant speed of 3 km/s from the Humps to Hundalee faults would take only half this duration. The geometrical segmentation of the Leader and Conway-Charwell faults delays rupture by more than 5 s. Rupture across the Conway-Charwell fault is initiated at shallow depth. The Stone Jug fault can subsequently only be activated after rupture reached the deep stress concentration area and unleashed its triggering potential, causing further delay.”

We stress the importance of this finding in the last section of our revised paper:

“We find that a zigzagged propagation path, accompanied by rupture delays at the transitions between faults, can explain apparently slow rupture speeds. While the surprisingly slow apparent rupture velocity and long rupture duration were depicted widely in seismological observations, our dynamic model provides a mechanically viable explanation for this observation.”

Finally, we add a sentence about this topic in the first paragraph of the introduction:

“The apparent rupture slowness is explained by a zigzagged propagation path and rupture delays at the transitions between faults.”

R1_4) Slip-weakening distance D_c at pag. 13 should be better introduced

D_c is now better introduced: “... a large slip-weakening distance D_c , the amount of slip over which frictional weakening occurs, has been estimated from a strong-motion record...”

R1_5) Typo:

Pag. 3 Ceska et al. → Cesca et al.

This is now corrected.

Reviewer #2 (Remarks to the Author):

Through a comprehensive, and well documented approach, the paper addresses an interesting question as to whether physics-based, dynamic rupture simulation can reproduce a set of observations associated with the complex Kaikōura (New Zealand) earthquake. The dynamic rupture model is based on a state-of-the-art numerical technique that includes realistic complexities, such as non-planar fault geometry, laboratory-derived friction laws, off-fault plasticity, material heterogeneities, and topography. This would be the first study to present dynamic rupture simulation of the complex Kaikōura earthquake. While findings about the rupture able to cascade without any contribution from the subduction interface are not able to rule the interface contribution totally out, they do bring an important conclusion from a dynamic perspective. However, I feel that three key issues need to be addressed to make this manuscript eligible for publication in Nature Communications: **broader objectives, improved data fits to support the model and concerns about the model being overly-tuned**. These three issues are described in more details below.

We believe we have thoroughly addressed all three key issues below.

General comment: the authors have covered the method in great details in the supplementary section, sometimes leaving the **main section too vague** (e.g., p8 “using static considerations, we first aim for optimal stress parameters within their identified uncertainties”).

We tried to avoid too technical discussions in the main text in an attempt to reach a broader audience than the dynamic earthquake rupture community. The reviewer’s comment suggests we erred too far in that direction. Also, our original manuscript had the very compact format of a Nature Geoscience paper. To take full advantage of the longer format allowed by Nature Communications, in the revised manuscript we have expanded the text. In particular, the paragraph which contains the sentence evoked by the reviewer has been rewritten as follow:

“Our initial stress model is fully described by seven independent parameters (fig. S6): four parameters related to regional stress and seismogenic depth, which are directly constrained by observations, and three unknown parameters related to fluid pressure, background shear stress and the intensity of deep stress concentration. A stress state is fully defined by its principal stress magnitudes and orientations. The orientations of all components and the relative magnitude of the intermediate principal stress are constrained by seismological observations (Townend et al., 2012). In addition, the smallest and largest principal stress

components are constrained by prescribing the prestress relatively to strength drop on optimally-oriented fault planes (Aochi and Madariaga, 2003). To determine the preferred initial stresses, we first ensure compatibility of the stress state with the prescribed fault geometry and the slip rakes inferred from static source inversion. In this purely static step, we determine optimal stress parameters, within their identified uncertainties, that maximize the ratio of shear to normal stress all over the fault and maximize the alignment between fault shear tractions and inferred slip (Xu et al., 2018). We then use a set of dynamic rupture simulations to determine the depth-dependent initial shear stress and fluid pressure that lead to subshear rupture and slip amounts consistent with previous source inversion studies. The resulting model incorporates over-pressurized fault zone fluids (Suppe, 2014; Sutherland et al., 2017; Uphoff et al., 2017) with a fluid pressure considerably higher than hydrostatic stress but well below lithostatic level (see method sec. A6).”

In addition, we moved most of section A3 (‘geometry of the fault system’) and fig. S1 of the original manuscript to the main text of the revised manuscript.

(I) Broader objectives

The study’s purpose is to unravel the “earthquake’s riddles in a physics based manner”, the 3 riddles being:

- a large gap separating fault segments according to Hamling et al., (2017) and Bradley et al. (2017),
- possibility of significant slip on the subduction interface
- and slow apparent rupture speed.

Regarding “the large gap”, it is worth noting that the Hamling et al. (2017) paper mentions “a large apparent gap”, as the fault are connected at depth in his model, and that the Bradley (2017) paper was a very preliminary attempt to obtain a kinematic model of the earthquake, however the waveform fits are not robustly supporting the suggested “20 sec” offset between ruptures.

The reviewer points out three riddles addressed by our model. Specifically, he notes that the reported surface rupture gap may not exist at depth. We are aware that the gap may be only apparent, as stated in the second paragraph of our submitted manuscript: “An apparent gap of 15-20 km between known fault structures (Hamling et al., 2017) may suggest a rupture jump over an unexpectedly large distance or the presence of deep fault segments connecting surface rupturing faults”. Our preliminary dynamic rupture experiments suggested that a distant triggering of the Upper Kowai fault (removed in our final model), for instance by the Hundalee segment rupture, is not feasible. On the other hand, our final model, connecting the fault in the hypocentral region with the Northernmost fault via the Point Kean fault, shows that the second hypothesis (connection by deep faults) is viable.

A ‘20 s offset’ between fault segments as proposed by Bradley et al. (2017) is indeed unlikely. On the other hand, Bradley et al. (2017)’s paper illustrates the difficulty of developing a model consistent with both geodetic and seismological data for this event. The key point we intend to make by citing all these unexpected reports, even if some were preliminary, is that “incorporating the requirement that the rupture should be dynamically viable can help constrain the unexpected features and competing views of this event” (as we stated in the introduction). Please note, that in motivating our modeling efforts, we did not consider the temporal gap of Bradley et al. (2017)’s model, but only the robust observation of a spatial gap of the surface rupture.

While the questions are of scientific interest, I feel that the driving motivation in terms of making a significant scientific and societal impact using advanced physics methods was not only to explain why the earthquake ruptured in such ways but also most importantly why it did not rupture at all as expected. Three more riddles should be investigated about the Kaikōura earthquake:

- rupturing exclusively to the North even though the recent Canterbury earthquakes significantly increased stress south of the epicentre,
- bypassing the Hope Fault, despite it being the major regional source of seismic hazard,
- and finally rupturing 200 km of crustal faults but stopping short just south of Wellington.

A suggested approach could be to first validate the model of the earthquake rupture and appropriate physics parameters, then to test/discuss these against highly expected rupture scenarios which however did not occur during the Kaikōura sequence.

We agree on the reviewer's suggestion to further investigate these questions, in addition to those in our original manuscript. Here are our findings concerning each of them:

1. Rupture to the South:

The recent Canterbury earthquakes produced negative and very modest Coulomb stress changes (0.02 MPa or less) in the hypocentral region of the Kaikōura earthquake and in the region directly South from the epicenter, where no rupture was observed (Fig R3, adapted from Steacy et al. (2013), illustrates this for the Darfield event, the largest earthquake of the Canterbury sequence).

Figure 2. Coulomb stress changes on 3-D optimally oriented planes from our initial slip model of the Darfield earthquake. The yellow line is the trace of the fault in that model, the yellow star the epicentre of the earthquake, and the yellow circles indicate $M \geq 5$ aftershocks that occurred after the Darfield earthquake and before the $M = 6.2$ Christchurch event. The epicentre of the latter is shown as a cyan star, that of the $M = 6.0$ June earthquake as a black star, and the December $M = 5.8$ and 5.9 events are indicated as blue and green stars, respectively. The open circles indicate the location of $M \geq 5$ aftershocks and the colours match those of the preceding large earthquakes.

Fig R3: Adapted from fig. 2 of Steacy et al. (2013), showing the Coulomb stress changes on 3-D optimally oriented planes produced by the Darfield earthquake. For spatial orientation, we here overlay the figure with a regional map. The red lines show the surface rupture trace of the Kaikoura event (Litchfield et al., 2018).

In our revised model, the geometry of the Humps segment to the Southwest has been slightly modified to better match the surface rupture mapping. The Southwestward rupture front on the new Humps segment now **stops spontaneously** (fig R4), which is explained by a less favorable orientation of this fault with respect to the regional tectonic stress state.

Fig R4: Spontaneous rupture termination to the SouthWest (left) of the Humps fault zone highlighted by rupture contours (every second) and by the final slip distribution.

From the New Zealand fault database, we identify two reverse faults which could have ruptured South of the Kaikoura event (fig R5): the Culverden Fault, dipping (SE) $30-70^\circ$ and 24 km long, and the Leonard Mound Fault, dipping (SE) $20-50^\circ$ and 22-24 km long (Pettinga et al., 2001). Among all possible dip angles, we choose the largest value, as it results in more loading under the assumed regional stress field. By including these faults (fig 1) in the dynamic rupture model presented in the manuscript, we now confirm that they would not break.

Fig R5: Map highlighting faults structures (Culverden and Leonard Mound faults) that could have ruptured South of the Kaikoura event (Langridge et al., 2016).

2. Bypassing the Hope Fault

As discussed in our answer to Reviewer 1, we updated our fault geometry in the hypocentral region. This refinement allowed us to reproduce the non-rupture of the Hope fault, explained by unfavourable dynamic stresses induced by the restraining step-over configuration of the Leader and Conway-Charwell faults.

3. Rupture termination to the North:

We wrote the following about this topic in our initial submission: “Finally, we locally reduce the initial stresses on the Northernmost part of the Needles fault in a way that mimics the stress shadow caused by the 2013 Cook Strait earthquake sequence (Hamling et al., 2014 and fig. S1). This prevents the occurrence of more than 10 m of fault slip in this area, which is not supported by inversion results (Hamling et al., 2017; Xu et al., 2018)”. In other words, a spontaneous rupture termination to the North is not achieved using our Needle fault

geometry under the adopted regional stress. Something else is needed to solve this riddle, for instance heterogeneities of initial stress or fault strength.

Following the reviewer's suggestion, we further investigated the rupture termination. The Needles fault is characterized by a very straight surface rupture (Litchfield et al., 2017). However, its geometry at depth is not well known. Hamling et al. (2017) and Wang et al. (2017) suggest that it dips 70° , whereas the geometry of Xu et al. (2018) features lower dip angles (30 to 50° depending on depth). The lateral spreading of the aftershocks in this region supports such low dip values but may also be related to the rupture of secondary faults, e.g. the Lighthouse fault, Cape Campbell Road fault and Marfells Beach fault. In any case, considering a steeper Needles fault would result in higher local shear stress. Bradley et al. (2017) noticed that the predominant wave packets at stations KEKS and WDFS are characterized by a large delay (about 16 s) despite their proximity (16 km separation), which they suggest might be caused by fault segmentation at the boundary between the Kekerengu and Needles faults. Thus, our geometry unifying Kekerengu and Needles faults may indeed be too simple. On the other hand, a high segmentation of the Needles fault is unlikely given its straight rupture trace, and cannot be invoked to explain the rupture termination. A more likely hypothesis could be the presence of an asperity in that region which terminated the rupture and which could be associated with the many localized aftershocks in the region. However, we removed our original argumentation of a stress shadow related to the 2013 Cook sequence being the cause of such heterogeneity (see details in a response at the end of this letter).

The following paragraph summing up these new results has been added to the discussion section:

“Features of the Kaikōura earthquake that remain unexplained by our dynamic models suggest opportunities to better understand the role of fault heterogeneities. These features include the inferred localized slip at depth on the Upper Kowhai fault as well as incompletely modeled aspects of the observed waveforms. Our dynamic rupture scenario is able to explain the early rupture termination to the South, but does not give a definitive answer concerning the origin of rupture termination to the North. On the Humps fault zone, spontaneous rupture termination to the West is observed, associated with a slight change in the strike direction resulting in a less favorable fault orientation. In additional dynamic rupture simulations including the nearest identified faults to the South, the Leonard Mound and the Culverden reverse faults (e.g. Pettinga et al., 2001), we found that the rupture is not able to trigger those faults. To the North, we have to locally reduce the initial stresses on the Northernmost part of the Needles fault to prevent its rupture with large slip. The very straight surface rupture of the Needles fault (Litchfield et al., 2017) does not suggest a high segmentation that may have prevented the rupture to extend further North. Hamling et al. (2017) and Wang et al. (2017) suggest a steeper geometry for this segment (dip angle of 70°) which would result in an increased shear over normal stress ratio, favouring rupture instead of terminating it. These considerations indicate that the most likely reason for the rupture termination to the North is the presence of an asperity to which the many aftershocks in the region (Kaiser et al., 2017) might be associated.”

(II) Improved data fits

While I appreciate that dynamic rupture model performance is in validating overall rupture processes, I have some concerns in the robustness of the results regarding quantitative comparison between modelled features and observed ones. **Figure 5** coseismic displacement shows **large discrepancies for the southernmost faults (EW and UD) and the Jordan and Papatea faults (opposite rupture mechanisms)**. **Figure 6** GPS vector displacements also shows **very large discrepancies with too much slip and inconsistent mechanism modelled** to the south and too little slip to the north of the Kaikōura rupture.

We address the concerns of the reviewer regarding robustness of our model by clarifying the scope of our dynamic, physics-based approach in distinction to kinematic, data-driven source modeling. We add on page 24:

“The physics-based dynamic source modeling approach in this study has distinct contributions compared to the data-driven kinematic source modeling approach. In the latter, a large number of free parameters enables close fitting of observations at the expense of mechanical consistency. Furthermore, the kinematic earthquake source inversion problem is inherently non-unique (many solutions fit the data equally well). In contrast, our dynamic model is controlled by a few independent parameters. Its main goal is to understand the underlying physics of the cascading rupture sequence. Adopting fault geometries and a regional stress state consistent with previous studies, our dynamic rupture model reproduces major observations of the real event, reveals unexpected features and constrains competing hypotheses.”

Moreover, there was a mistake in our script to compute the residual ground deformation in figure 5 of the original manuscript, which gave the incorrect impression of a poor agreement between model and observations. Previously, we computed the residual by:

$$\text{residual} = \text{observation} - (\text{synthetics}(110\text{s}) + \text{synthetics}(40\text{s}))$$

Our script added the data “synthetics(40s)” based on a specific simulation which took into account a restart of the SeisSol code using a checkpoint. For the following simulations that ran through without checkpointing we forgot to remove this term. This led to wrong, large residuals around the Southern parts of the rupture (fig R6). This bug is now fixed and the corrected figure is reproduced below.

Fig R6: Comparison of residual ground displacements (observed - synthetics) for the revised model with (above) and without (below) the bug in the residual calculation.

The fault geometry underlying our models has now been refined in the hypocentral region using recent fault trace and fault scarps data in that region (Nicol et al., 2018, see also answer to Reviewer 1). A slight change in the western part of the Humps fault, a steeper fault geometry to the North of the new Leader and Conway-Charwell faults and a steeper Stone Jug fault, allow to match the observed ground-deformations more accurately (figs. 6 and 7 of the revised manuscript).

Finally, the mechanisms of our original model were in our opinion not inconsistent as criticized by the reviewer. Our modeled rupture mechanisms on the Jordan and Papatea faults are in agreement with the inferred slip rake angles of Xu et al. (2018)'s, which in turn reproduce the static ground-deformation well.

With regard to the proposed model time history evolution, Figure S4 synthetic waveforms for station KEKS shows over 10 second delay with actual recorded waveforms, therefore opening some concerns about the overall robustness of the proposed dynamic model. These points are described in more detail below.

+ While I understand that the dynamic rupture model is tuned to fit the source time function estimated from published kinematic models, **removing the Upper Kowhai fault out of the fault model cannot be justified**, as there is strong geodetic evidence suggesting that a moderate amount of slip at depths on Upper Kowhai occurred during the Kaikōura earthquake (Hamling et al., 2017; Xu et al., 2017).

We justify our adopted Upper Kowhai geometry in a detailed answer to reviewer 1, above. We added a related discussion in section 1.1 of the revised paper.

This type of consideration should be motivated by the data. Local strong-motion and high-rate GPS data (both available via GeoNet: <https://www.geonet.org.nz/>) **do not really support the sequential rupture from Point Kean, to Papatea, to Kekerengu**. In addition, Figure 6 shows a significant disagreement between the model and GPS data around the town of Kaikōura and the Upper Kowhai fault.

We provide in this rebuttal letter new comparisons between modelled and observed strong motion, GPS and teleseismic data. Specifically, we show that the ground motions recorded at station KEKS are consistent with a sequential rupture from the Papatea to Kekerengu faults. Such sequential rupture suggests that the Papatea fault was triggered at its Southern

end. The Point Kean fault, as the nearest fault, then appears as the most likely cause for such triggering.

We are aware of the significant disagreement between the model and GPS data around the town of Kaikōura. This question remains open and would have to be addressed by locally refined fault information or heterogeneities, as we already commented in the original manuscript: “On the other hand, a stronger dip-slip component would be required to explain the northeastward GPS displacements around this thrust fault. According to the dynamic rupture model, this could only be achieved by an (unlikely) local prestress rotation of about 30 degrees towards South, or by considering a fault geometry with lower strike.”

By the way, please provide root-mean-square misfit or variance reduction, as the fits shown in Figure 6, especially the vertical component, appear to be not good. Also, please move Figure S1 to the main paper, which nicely highlights several key assumptions in this study.

RMS misfit has been added to fig 6 and 7 (of the revised manuscript) and figure S1 was moved to the main paper (now figure 2 of the revised manuscript).

+ Furthermore, published papers have not yet shown any **evidence that the Papatea fault was ruptured** before the Kekerengu fault. If anything, this is the other way around.

Strong evidence for a rupture sequence from Papatea to Kekerengu is further provided by the teleseismic back-projection results of Xu et al (2018). More recently, comparing remote sensing and field observations to 2D dynamic simulation results, Klinger et al. (2018) showed that observed patterns of surface slip and off-fault damage support this scenario.

Comparison of synthetic seismograms at station KEKS (Figure S4) with the data presented in published papers suggests a significant time shift (of more than 10 seconds) between the arrivals of the dominant phases at station KEKS, indicating that the model is not consistent with local seismic data. If the authors simply shift the rupture time of the Kekerengu fault to match the seismic data, then the source time function would become incompatible with those estimated from kinematic rupture models (Figure 4f). Please discuss these caveats and carefully interpret the modeling result.

First of all, we corrected an unfortunate mistake in plotting the synthetic KEKS record in Fig. S4. We forgot to first rotate the records from the station reference frame to the ENZ reference frame. Fig S4 (now S3) has been updated. Our updated analysis, describe below, shows that our model captures the timing of waveforms sufficiently well, apart from a small, consistent delay of first arrivals probably associated with a too rapid nucleation procedure. Specifically, we can dispel the suspicion of the reviewer of an overall 10 seconds time shift of the dominant waveform pulses at KEKS, which was spuriously induced by our flawed figure S4.

We agree that strong ground motions and continuous GPS data provide valuable additional constraints on the rupture kinematics. After our original submission, C. Holden and E. D’Anastasio kindly provided us processed GPS time histories that we can compare to our synthetics. Newly performed comparison to strong motion, GPS and teleseismic observations are now added to the results section of the manuscript (page 22 ff.), including two new figures (figs. 8 and 9) as detailed in the following.

“Strong ground motion and continuous GPS data provide valuable constraints on the rupture kinematics. We compare our simulation results to these data with a focus on the timing of pulses, because our model does not account for small scale heterogeneities which could significantly modulate waveforms. Due to the close distance of some of the stations to the faults (fig. 8) a close match of synthetic and observed waveforms is not expected. Yet, the dynamic rupture model is able to reproduce key features of the strong ground-motion and GPS recordings (fig. 9). Our model captures the shape and amplitude of some pronounced waveform pulses, e. g. of the first strong pulse recorded along the NS direction at GPS station MRBL, which is situated in the nucleation area. A time shift of around 2 s hints at a nucleation process slower than modeled.”

In this rebuttal letter we now focus on some of the near-fault stations, from South-West to North-East: MRBL, KAIK, KIKS, KEKS and CMBL.

The KAIK, KIKS and KEKS stations are extremely close to ruptured faults: we expect at these stations artifacts due to our use of a simplified fault geometry. Synthetic first arrivals at station KAIK also appear too early by around 3 s. The rupture of the northern part of the Hundalee fault produces strong signatures in the synthetics, around $t=36$ s, whereas a similar phase is not clearly visible in the observed records. Our modeled synthetics could be improved by detailed geometry and/or asperity information about the Northern part of the Hundalee fault (White faults). However, the expected effects on the dynamic rupture cascade are marginal.

At station KIKS, two dominant phases visible on the NS observed records between 25 s and 45 s roughly coincide with peaks of the synthetics waveforms. The modeled waveforms appear again ahead of time by around 2 s.

We now discuss the KEKS synthetics in detail:

The dynamic rupture front directly passing by station KEKS between 64 and 70 s leaves a clear signature featuring high amplitudes in the synthetics. Quite interestingly, such a peak, of much smaller amplitude and larger duration than modeled, is also visible in the observed records (e.g. NS component) between 62 and 73 s. The large modeled amplitude at KEKS may be related to an overly smooth geometry of the Kekerengu fault, which favours the generation of extreme, very coherent, ground motions. Accounting for small-scale roughness of the otherwise straight fault geometry may result in broadening and amplitude reduction of the generated strong motions.

Displacement waveforms at station CMBL, further North, show rather good agreement in terms of timing and rise-time, supporting our dynamic rupture scenario.

In the revised manuscript we discuss details of KEKS synthetics important for the dynamic rupture cascade as follows:

“At near-fault station KEKS two dominant phases are visible on both observed and synthetics waveforms (at 52 s and 63 s after rupture onset in the NS synthetics of fig. 9 and in the fault-parallel-rotated waveforms of fig. S3). These dominant phases were attributed to a slip reactivation process on the Kekerengu fault by Holden et al. (2017). However, our model suggests that the first peak stems from the earlier rupture of the Papatea segment (see animation A1.B). The ground motions recorded at station KEKS are thus consistent with a sequential rupture from the Papatea to Kekerengu faults.”

+ The authors imply crustal faults ruptured during the Kaikōura earthquake are weak, citing evidence that crustal faults in the Marlborough region are not optimally oriented with respect to the regional stress inferred from seismological observations (Townend et al. 2012). Townend et al.'s inference was for major Marlborough faults (e.g., Hope, Clarence, Awatere, and Wairau) that are vertical strike-slip. In contrast, many faults (Hundalee, Jordan, Kekerengu, Needles, etc) ruptured during the Kaikōura quake dip toward northwest (50 – 70 degrees) and hence are more or less optimally oriented with respect to the regional stress field. Therefore I am puzzled by the authors' claim about the weakness of these faults. **I would argue that the weakness of these crustal faults is simply the outcome of the assumed friction laws (not the outcome of the modeling result), in that in order to produce reasonable stress drop with enhance dynamic weakening, the prestress needs to be quite low (e.g., Noda et al., 2009).**

We wrote in the introduction: "Much like the San Andreas fault, the Marlborough fault system is apparently weak, according to its large angle relative to the maximum horizontal compressive stress (Townend et al., 2012)". Generalizing this statement to all the faults that ruptured in the Kaikoura earthquake is indeed wrong. The reviewers arguments prompted us to clearly formulate our model assumptions in terms of fault 'weakness', which led to a considerable extension of the methods section A7 entitled 'Apparent fault weakness'. Therein we now discuss the orientation of the fault system, apparent fault strengths in the static and dynamic sense and explore additional model setups probing the robustness of our preferred parametrization. We conclude by quantifying the relative contributions of each modeling assumption to the apparent weakness of faults in our model.

In addition, we change the following parts of the manuscript:

We remove the reference to the San Andreas fault in the introduction, and rewrote the paragraph as follows:

"Mature plate boundary faults are, in general, apparently weak (Zoback et al., 1987; Behr and Platt, 2014; England, 2018), a feature that is required also by long-term geodynamic processes (e.g. Duarte et al., 2015; Osei Tutu et al., 2018) but that seems incompatible with the high static frictional strength of rocks (Byerlee, 1978). These two observations can be reconciled by considering dynamic weakening, which allows faults to operate at low average shear stress (Noda et al., 2009). However, low background stresses are generally unfavourable for rupture cascading across a network of faults. For instance, rupture jumps across fault stepovers are hindered by low initial stresses (Bai and Ampuero, 2017). This is one reason why finding a viable dynamic rupture model is non-trivial. The modelled fault system presented here features a low apparent friction while being overall favourably oriented with respect to the background stress. We demonstrate that fault weakness is compatible with a multi-fault cascading rupture. Our models suggest that such a weak-fault state is actually required to reproduce the Kaikoura cascade (see method sec. A7)."

We also restructured and clarified the corresponding paragraphs of section 1.3:

"In our model, dynamic rupture cascading is facilitated by deep stress concentrations (Fig. 2). The presence of stress concentrations at depth near the rheological transition between the locked and steady sliding portions of a fault is a known mathematical result of the theory of dislocations in elastic media (e.g. Kato, 2012, Bruhat and Segall, 2017). Such stress concentrations are also a typical result of interseismic stress calculations based on

geodetically-derived coupling maps (Ader et al., 2012) or long-term slip rates (Mildon et al., 2017). Stress concentrations due to deep creep on the megathrust have been proposed to determine the rupture path independent of crustal fault characteristics (Lamb et al. 2018). Stress concentration is introduced in our model by two independent modulation functions (Fig. S8).

Our initial stress model leads to low values of the initial shear to normal stress ratio over most of the seismogenic zone (the median value over the rupture area is 0.09, see Fig. S9) in consistence with the apparent weakness of faults (Copley, 2018, methods section A7). Yet, most faults of our model are relatively well oriented with respect to the regional stress, and are therefore not weak in the classical sense. The classical Andersonian theory of faulting may be challenged in transpressional tectonic stress regimes resulting in non-unique faulting mechanisms. In the framework of dynamic rupture modeling, faults can be stressed well below failure almost everywhere and yet break spontaneously if triggered by a small highly stressed patch. Under the assumption of severe velocity-weakening friction (detailed in the previous section), a low level of prestress is required to achieve a reasonable stress drop. To this end, we have considered here two effects rarely taken into account together in dynamic rupture scenarios: 1) increased fluid pressure and 2) deep stress concentrations. We discuss their trade offs in more detail in section A6. We infer that the interplay of deep creep, elevated fluid pressure and frictional dynamic weakening govern the apparent strength of faults and that these factors cannot be treated in isolation for such complex fault systems.”

We now review our assumption of $f_w=0.1$ in the discussion section, within the paragraph about ‘fault weakness’:

“Our results provide insight on the state of stress under which complex fault systems operate. In our model, strong frictional weakening, fluid overpressure and deep stress concentrations result in a remarkably low apparent friction. Yet the low average ratio of initial shear stress to normal stress does not hinder dynamic rupture cascading across multiple fault segments. Instead, it is crucial to achieve the full cascading rupture with realistic stress drop and slip. In methods section A7 we discuss fault strength based on the orientation of the fault system, apparent fault strengths in the static and dynamic sense and explore additional model setups demonstrating the robustness of our preferred model. We conclude by quantifying the relative contributions of our modeling assumptions to the apparent weakness of faults.

The effects of overpressurized fault fluids and deep stress concentrations and the additional effect of a low dynamic friction result in an overall low apparent friction coefficient. We find that reproducing all aspects of the rupture cascade requires all three effects. The combined effect of strong frictional weakening, fluid overpressure and deep stress concentrations and the fundamental impact of fault weakness on the existence of subduction and tectonics (e.g. Osei Tutu et al., 2018) show the importance of mechanical feedbacks across multiple time scales, from the short-term processes of dynamic rupture and earthquake cycles to the long-term geodynamic processes that shape and reshape the Earth.

Dramatic frictional weakening is one of the key mechanisms contributing to fault weakness in our model. Our assumed dynamic friction coefficient, $f_w=0.1$, falls within the range of values

typically observed in laboratory experiments and considered by the dynamic rupture community (e.g. Noda et al., 2009; Gabriel et al., 2012; Shi and Day, 2013). Nevertheless, we probed the necessity of such a low value by additional simulations, as detailed in methods sec. A7d. By static considerations, we find that a sustained cascading rupture under a higher f_w would require conditions that disagree with stress inversion inferences, namely a too low stress shape ratio (Fig. S5). In addition, prescribing higher f_w results in a prestress distribution of larger variability, less favourable for rupture cascading. “

Finally, we add a sentence about this topic in the abstract:

“The complex fault system operates at low apparent friction thanks to the combined effects of overpressurized fluids, low dynamic friction and stress concentrations induced by deep fault creep.”

(III) Model over-tuned

As a result of the complicated modeling approach, it is difficult to understand the **robustness of the claims made in this study and sensitivity of the results to underlying assumptions**. My impression is that the model is overly tuned to match observations and kinematic rupture models.

We included three fault weakness ingredients based on hypothesis backed up by observations and laboratory experiments. We discussed their trade offs quantitatively in the original methods section A7 ‘Initial Stresses’ and A8 ‘Apparent fault weakness’ (which is considerably extended in the revised version). We constrain all modeling ingredients by physical and observational constraints. We nevertheless acknowledge the possibility of alternative models yielding similar rupture dynamics. Such models can be readily designed based on the trade-offs we define in methods section A6, e. g. by decreasing or increasing the effects of deep stress concentrations, fluid pressure or frictional weakening. We conducted additional dynamic rupture simulations to address this comment. We summarize two of those (DR2 and DR3) in the new subsection A7.d ‘Alternative model setups featuring higher dynamic friction coefficients to probe the robustness of our preferred model’. However, none of the models we explored performed equally well as the preferred model achieved by following the steps outlined in Fig. S6.

+ The authors assume a particular shape of the depth-dependence stress functions shown in Figure S9, which are rather arbitrary aside from the fact that some increase of stress at depths near the rheological transition is intuitively expected. **Why not simply assume a linear increase with depth?** It looks to me that the exact shape of these functions strongly controls the rupture propagation behavior (Figure S1). Additional simulations and discussion would be needed to quantify the sensitivity of these assumed functions to the resulting rupture behavior.

In addition to the two dynamic rupture simulations with varied f_w discussed in methods section A7 of the revised manuscript, we specifically address this comment by performing a new model DR1 comparable to our preferred model but omitting deep stress concentrations. We assume a linear increase of stress functions with depth as suggested by the reviewer. We added a discussion of our findings to the manuscript in methods section A6:

“To probe the importance of deep stress concentration, we performed a new model DR1 comparable to our preferred model but omitting deep stress concentrations. We decrease R_0 and simultaneously adjust the fluid pressure ratio γ to preserve the average stress drop, and find the smallest R_0 enabling the full rupture cascade. The model DR1 has $R_0 = R_{opt}(z) =$

0.7 and $\gamma = 0.7$ (Table 2). Its final fault slip is roughly similar to the slip of our preferred model. However, this alternative model has drawbacks compared to observations. In particular, it is less realistic in terms of timing. Its overall rupture duration is about 10 s shorter than our best scenario. This difference is mainly due to quicker shallow rupture transitions, such as the Humps-Leader branching, which are made easier by the increased prestress at shallow depth. Although this alternative model does not compare as well with observations as our preferred model, we cannot exclude the existence of an equally well performing model featuring less pronounced stress concentrations.”. Such model could be achieved for instance by including stochastic small scale heterogeneities or fault roughness. However such considerations are beyond the scope of the current paper, as we note throughout the manuscript (e.g. in the discussion section).

While linearly depth-dependent stress is a standard assumption in dynamic rupture modeling, we motivate the assumption of deep stress concentrations in our model in the (now slightly extended) section 1.3 of the manuscript :

“In our model, dynamic rupture cascading is facilitated by deep stress concentrations (Fig. 2). The presence of stress concentrations at depth near the rheological transition between the locked and steady sliding portions of a fault is a known mathematical result of the theory of dislocations in elastic media (e.g. Kato, 2012, Bruhat and Segall, 2017). Such stress concentrations are also a typical result of interseismic stress calculations based on geodetically-derived coupling maps (Ader et al., 2012) or long-term slip rates (Mildon et al, 2017). Stress concentrations due to deep creep on the megathrust have been proposed to determine the rupture path independent of crustal fault characteristics (Lamb et al. 2018). Stress concentration is introduced in our model by two independent modulation functions (Fig. S8).”

Those results do not predict a linear increase with depth of the interseismic stressing rate, but rather a stress concentration at depth. We further write in methods section A6:

“Our stress modulation function is described by a minimum number of parameters (the width of the stress concentration area, the seismogenic depth z_{seis} and the stress concentration shape, described hereafter). It is designed to capture the essential features of the stresses caused by deep creep: it is peaked at the base of the seismogenic zone ($g(z)=1$) and decays to $g(0) < 1$ at shallower depth to represent the background stress. Most probably, any function with these general features could be used to achieve similar dynamic rupture results.”

As suggested by our new simulations, the shape of the depth-dependency of stress used in our model is probably non-unique, and should be only seen as a conceptual model we assume realistic. In future work, the depth-dependency of stress could be constrained by seismic cycle modeling capable of handling complex fault geometries.

In section A8 of the original manuscript (now A6) we comment on the physical and observational constraints of the modulation functions we introduce, namely stress drop and slip distribution.

+ On reducing initial stress on the Needles fault. If stress changes on the Needles fault induced by the 2013 Cook Strait sequences are used to justify the removal of 10 m coseismic slip, there is something wrong with the estimate of the stress changes. Note that the Cook Strait and Lake Grassmere earthquakes are M6.5-6.6, with the maximum slip of the order of 3 m, which would have induced relatively small stress changes (<0.5 MPa) on the Needles fault.

We have now explicitly computed the stress changes induced by the 2013 Cook Strait sequences (fig R7 and R8) and it turns out that they are too small to justify a strong decrease of initial stress (R_0) in that region. Such initial stress heterogeneity is nevertheless needed to stop the rupture. As discussed in section 3 of the original manuscript, the origin of rupture termination to the North remains a non-trivial puzzle from the perspective of dynamics.

In the revised manuscript, the idea that the stress decrease is caused by the stress shadow of the 2013 Cook Strait earthquake sequence has been removed. We now write:

“Finally, we locally reduce the initial stresses on the Northernmost part of the Needles fault to prevent the occurrence of large slip in this area. We find that the Needles fault would otherwise host more than 10 m of slip, which is not supported by inversion results (Hamling et al., 2017; Xu et al., 2018).”

The inferred slip of those earthquakes has also been removed from figure S1 (now named figure 2).

Fig R7: Stress changes caused by the Cook Strait and Lake Grassmere earthquakes. T_s and T_d are the shear tractions along strike and dip and P_n is the normal stress. The inferred tractions are noisy at the intersection of the Cook Strait and Needles faults because the events are imposed as point sources. Nevertheless, this does not prevent estimating the stress changes further away from the intersections.

Fig R8: left: Distribution of initial fault prestress ratio R in the Needles fault region, if a laterally homogeneous R_0 is imposed. Right: Distribution of initial fault prestress ratio R accounting for the 2013 Cook Strait sequence. The R ratio remains in fact mostly unaffected by the earthquake sequence as pointed out by Reviewer 2.

References:

Langridge, R. M., et al. "The New Zealand active faults database." *New Zealand Journal of Geology and Geophysics* 59.1 (2016): 86-96.

Litchfield, N. J., et al. (2018). Surface Rupture of Multiple Crustal Faults in the 2016 M w 7.8 Kaikōura, New Zealand, Earthquake. *Bulletin of the Seismological Society of America*.

Steady, S., Jiménez, A., & Holden, C. (2013). Stress triggering and the Canterbury earthquake sequence. *Geophysical Journal International*, 196(1), 473-480.

REVIEWERS' COMMENTS:

Reviewer #1 (Remarks to the Author):

This is a nice, new contribution on the rupture dynamics of the Kaikoura earthquake, which showed an interesting and complex rupture process at a subduction termination. Beside the available literature on this earthquake, this study provides new and interesting results, in particular with respect to the presented dynamic model, able to explain the complex cascade rupture on multiple faults and slow apparent rupture duration. Authors carefully considered all reviewers suggestions. I think the paper can now be accepted for publication.
Simone Cesca

Reviewer #2 (Remarks to the Author):

Thank you for thoroughly reviewing the manuscript. I am very pleased with the corrections/added information and appreciate your efforts put in comparing regional fits and especially teleseismic fits. I only have minor comments:

- please spell Kaikōura consistently throughout the manuscript
- Some figures require a distance scale and possibly orientation (Fig. 1, 2,3 etc)

Reviewer #1 (Remarks to the Author):

This is a nice, new contribution on the rupture dynamics of the Kaikoura earthquake, which showed an interesting and complex rupture process at a subduction termination. Beside the available literature on this earthquake, this study provides new and interesting results, in particular with respect to the presented dynamic model, able to explain the complex cascade rupture on multiple faults and slow apparent rupture duration. Authors carefully considered all reviewers suggestions. I think the paper can now be accepted for publication.
Simone Cesca

Reviewer #2 (Remarks to the Author):

Thank you for thoroughly reviewing the manuscript. I am very pleased with the corrections/added information and appreciate your efforts put in comparing regional fits and especially teleseismic fits. I only have minor comments:
- please spell Kaikōura consistently throughout the manuscript
- Some figures require a distance scale and possibly orientation (Fig. 1, 2,3 etc)

Kaikōura is now spelled consistently throughout the manuscript.

The North direction and a distance scale have been added to figure 2. Latitude and longitude lines have been added to figures 1 and 3.

Apart from the above, we report the following changes:

- The synthetic displacement presented in figure 4 were from a outdated simulation. We fixed this mistake.
- The teleseismic comparisons are now computed using a Green's function database accurate to 2 s period (previously 10 s).
- The code of the projection we use was wrong in the text. This has been corrected.
- Subheaders have been added to the results section
- Following discussion with the Editor, we now integrate the section Model into section Results (sections Fault geometry, Friction and Initial stress have been moved to Results). Section 'Numerical methods' has been moved to Methods. The reference to Figure 3 has been moved from Numerical methods to Results, to keep this figure in the main text.
- We rewrote the first and last paragraph of the introduction.